# A Comprehensive Review of Remote Sensing and Artificial Intelligence Integration: Advances, Applications, and Challenges

**DOI:** 10.3390/s25195965

**Published:** 2025-09-25

**Authors:** Nikolay Kazanskiy, Roman Khabibullin, Artem Nikonorov, Svetlana Khonina

**Affiliations:** Samara National Research University, Samara 443086, Russia; kazanskiy@ssau.ru (N.K.); nikonorov.av@ssau.ru (A.N.); khonina.sn@ssau.ru (S.K.)

**Keywords:** remote sensing, artificial intelligence, machine learning, deep learning, vegetation mapping, explainable AI (XAI), multispectral data

## Abstract

The integration of remote sensing (RS) and artificial intelligence (AI) has revolutionized Earth observation, enabling automated, efficient, and precise analysis of vast and complex datasets. RS techniques, leveraging satellite imagery, aerial photography, and ground-based sensors, provide critical insights into environmental monitoring, disaster response, agriculture, and urban planning. The rapid developments in AI, specifically machine learning (ML) and deep learning (DL), have significantly enhanced the processing and interpretation of RS data. AI-powered models, including convolutional neural networks (CNNs), recurrent neural networks (RNNs), and reinforcement learning (RL) algorithms, have demonstrated remarkable capabilities in feature extraction, classification, anomaly detection, and predictive modeling. This paper provides a comprehensive survey of the latest developments at the intersection of RS and AI, highlighting key methodologies, applications, and emerging challenges. While AI-driven RS offers unprecedented opportunities for automation and decision-making, issues related to model generalization, explainability, data heterogeneity, and ethical considerations remain significant hurdles. The review concludes by discussing future research directions, emphasizing the need for improved model interpretability, multimodal learning, and real-time AI deployment for global-scale applications.

## 1. Introduction

Remote sensing (RS), the practice of collecting information about the Earth’s surface without direct contact, has revolutionized how we monitor, analyze, and manage our planet’s dynamic systems [1,2]. With the increasing availability of satellite imagery, aerial photography, and ground-based sensor data, RS has become a cornerstone for addressing a broad range of scientific and societal challenges, from environmental monitoring and disaster management to urban planning and agriculture [3,4,5]. However, the sheer volume, complexity, and diversity of data produced by modern RS technologies present significant challenges in data processing, analysis, and interpretation [6].

In parallel, artificial intelligence (AI), particularly the advancements in machine learning (ML) and deep learning (DL), has exhibited extraordinary capabilities in extracting meaningful patterns from complex datasets [7,8]. AI offers sophisticated algorithms capable of handling high-dimensional data, recognizing intricate patterns, and adapting to diverse problem domains with minimal human intervention [9,10]. These attributes make AI an ideal candidate for tackling the challenges inherent in RS data analysis, including feature extraction, image classification, object detection, and time-series analysis [11,12].

RS leverages a variety of algorithms and AI models to process, analyze, and interpret data collected from satellites, drones, and other platforms. ML models, such as Support Vector Machines (SVM) [13] and Random Forests (RFs) [14], are commonly used for tasks like land cover classification and anomaly detection. DL models, especially CNNs [15,16], excel in extracting spatial features from high-resolution imagery for applications like object detection, change detection, and segmentation. RNNs [17] and Long Short-Term Memory (LSTM) networks are applied to analyze time-series data [18], such as vegetation dynamics or climate patterns. Advanced algorithms like Principal Component Analysis (PCA) aid in dimensionality reduction [19], while clustering techniques like K-Means and Density-based spatial clustering of applications with noise (DBSCAN) help in unsupervised classification [20]. AI-powered techniques streamline the interpretation of massive datasets, enabling more accurate and efficient insights for environmental monitoring, agriculture, urban planning, and disaster management.

The convergence of RS and AI has opened new avenues for innovation, enabling more accurate, efficient, and automated solutions across various application areas [21]. For instance, AI-driven techniques have substantially boosted the precision of land cover classification, improved the timeliness of disaster response, and expanded the scope of biodiversity monitoring [22]. Furthermore, these advancements are underpinned by a growing ecosystem of accessible datasets, pre-trained models, and computational resources, which have democratized the adoption of AI methods in the RS community [23]. Despite these promising developments, integrating RS and AI also introduces new challenges [24]. Issues such as model generalization across diverse geographic regions, the interpretability of AI models, and the ethical implications of automated decision-making require careful consideration [7]. Moreover, bridging the gap between domain expertise in RS and technical proficiency in AI remains a critical bottleneck for broader adoption and success [25].

Although several surveys have reviewed ML or DL in RS, most existing works remain method-specific or application-specific, resulting in a fragmented understanding of the field. Prior reviews are often descriptive and rarely provide a critical evaluation of the comparative strengths, limitations, and suitability of different approaches across diverse RS tasks. Few surveys integrate classical ML, modern DL, and emerging AI techniques (such as transformers and GANs) within a unified framework while also illustrating their use through representative case studies. This review addresses these gaps by offering a comprehensive and critical synthesis that spans the AI spectrum, incorporates real-world RS applications, and highlights methodological challenges alongside prospects. In doing so, it complements and extends earlier surveys at a time when AI in RS is evolving rapidly. To ensure a structured and reproducible review, we conducted a literature search across Scopus, Web of Science, IEEE Xplore, and Google Scholar up to June 2025. The keywords combined terms related to artificial intelligence (AI), machine learning (ML), and remote sensing (RS), including “AI in remote sensing,” “machine learning classification,” “deep learning satellite imagery,” “land cover detection,” and “environmental monitoring.” We included peer-reviewed journal articles and conference proceedings published in English between 2000 and 2025 that applied AI/ML techniques to RS data. Studies were excluded if they (1) focused solely on algorithm development without RS applications, (2) were review/commentary pieces, or (3) lacked methodological detail relevant to RS applications. This process ensured that the review covered both methodological innovations and real-world applications in environmental monitoring and land cover analysis.

Remote sensing has advanced rapidly in recent decades, producing large volumes of multispectral, hyperspectral, radar, and LiDAR data. Traditional processing methods often struggle to capture non-linear relationships or to scale efficiently with these massive and heterogeneous datasets. AI, encompassing both classical machine learning (ML) and modern deep learning (DL), offers powerful alternatives for feature extraction, classification, and prediction.

In this review, ML and DL are treated in detail in Section 3.2 and Section 3.3, respectively. Their inclusion reflects the growing consensus that AI methods are no longer supplementary, but rather central to advancing the state of the art in RS. By highlighting both methodological advances and practical case studies, this review aims to show how AI is transforming applications ranging from disaster response to agriculture and biodiversity monitoring.

## 2. Fundamentals of RS

RS is a technology used to acquire information about objects or areas from a distance, without direct physical contact. It detects and measures electromagnetic radiation that is either naturally emitted, reflected or scattered by the Earth’s surface or atmosphere. The fundamental principle involves capturing energy across different regions of the electromagnetic spectrum, such as visible light, infrared, microwave, and ultraviolet, using sensors attached on platforms like satellites, aircraft, drones, or ground-based systems [6]. There are two main types of RS: passive and active. Passive systems, such as optical sensors, rely on external energy sources like sunlight. For instance, a satellite equipped with a multispectral sensor measures the sunlight reflected by vegetation to assess its health based on specific wavelength absorption patterns, such as in the visible and NIR regions. Active RS systems, such as radar or Light Detection and Ranging (LiDAR), generate their own energy by emitting pulses of microwave or laser beams toward the target and analyzing the returned signal [26,27]. An example of this is LiDAR being used to map forest canopies or urban infrastructure with high precision by measuring the time delay and intensity of the reflected laser signals [28,29,30].

The application of RS is vast and multidisciplinary. For instance, in agriculture, RS helps monitor crop health, soil moisture levels, and pest infestations using indices like the Normalized Difference Vegetation Index (NDVI) [31]. In environmental studies, it aids in tracking deforestation, melting glaciers, and the spread of wildfires, often using satellite imagery from platforms like Landsat or Sentinel [32,33,34]. During natural disasters, RS supports real-time monitoring and response by providing critical information on flood extent, earthquake damage, or storm trajectories, as seen in data supplied by radar satellites like Sentinel-1 during floods [35]. Garg et al. explored the potential of Sentinel-1 SAR for near-real-time flood mapping in arid regions [35]. Key parameters—polarization, temporal changes, and interferometric coherence—were analyzed across flood events in Iran, Pakistan, and Turkmenistan using RF models. Integrating VV coherence and amplitude from pre- and post-flood imagery proved most effective, reducing computational time by 35% and improving accuracy by 50%. Advances in cloud processing have overcome traditional computational barriers, making this adaptable approach a step forward for global flood mapping in arid regions.

RS also plays a pivotal role in urban planning, allowing for the study of urban heat islands, infrastructure development, and land-use changes over time. Advanced technologies, such as hyperspectral imaging (HSI), further enhance the capability of RS by acquiring hundreds of narrow spectral bands, enabling detailed material identification, such as distinguishing different mineral compositions. Atmospheric conditions, spatial resolution, temporal frequency, and sensor calibration are critical factors influencing the effectiveness and accuracy of RS data. By integrating RS data with geographic information systems (GIS) [36,37], researchers and decision-makers can create actionable insights for sustainable development, resource management, and disaster mitigation, demonstrating this technology’s transformative potential.

Xia et al. integrated RS and geolocation datasets to extract urban areas across China [38]. Features such as night-time lights, vegetation cover, land surface temperature, population density, LRD, and road networks were analyzed using a RF classifier. Labeled samples from high-resolution land cover products and Landsat data were used for training and validation. The method identified 176,266 km^2^ of urban areas with 90.79% accuracy and a kappa coefficient of 0.790. Night-time lights and LRD contributed most to predictions, and results aligned closely with manually interpreted urban boundaries in selected cities. The findings demonstrated the potential of integrating RS and geolocation data for large-scale urban mapping, offering a reliable solution for monitoring urbanization.

As a case study, Stockholm has been used to illustrate the fundamentals of RS. The study applied multispectral data to analyze land cover dynamics, focusing on how spectral indices differentiate urban structures, vegetation, and water. The main finding was that RS techniques effectively captured spatial variations in an urban setting, demonstrating the value of spectral information for classification and monitoring. Figure 1 provides a visual overview of the study area and supports the explanation of basic RS principles.

## 3. Integration of AI, ML, and DL in RS

This section reviews the integration of AI in remote sensing. We begin with representative case studies (Section 3.1) that illustrate how AI-powered methods are already applied across domains such as disaster response, agriculture, and biodiversity monitoring. These examples highlight the practical impact of AI across diverse RS tasks and provide cross-cutting applications that contextualize the methodological discussions on classical ML approaches (Section 3.2) and modern DL models (Section 3.3).

### 3.1. AI in RS: Illustrative Case Studies

AI plays a pivotal role in RS by enabling advanced data analysis and interpretation, revolutionizing the way we understand and utilize data from satellites and other sensors. Through the integration of sophisticated algorithms, AI empowers the extraction of meaningful and actionable insights from vast and complex datasets, often far beyond human capabilities [40,41]. Techniques like computer vision and natural language processing (NLP) are at the forefront of these advancements, with computer vision excelling in analyzing visual data, such as satellite images, and NLP aiding in processing and understanding metadata or textual descriptions associated with the imagery [42,43].

In RS, AI algorithms are deployed for a range of critical tasks, including classification, segmentation, anomaly detection, and predictive modeling. Classification helps identify and categorize different types of land cover, vegetation, or water bodies, while segmentation focuses on partitioning images into distinct regions, such as urban, agricultural, or forested areas. Anomaly detection plays a crucial role in identifying unusual or unexpected changes, such as the emergence of illegal mining activities, deforestation, or natural disasters like landslides [44]. Predictive modeling, on the other hand, leverages historical and real-time data to forecast future events, such as crop yields or the progression of urban development.

The types of AI used in RS are diverse, with expert systems, neural networks, and RL each bringing unique strengths [8,11]. Expert systems rely on pre-defined rules and domain knowledge to make decisions, ideal for tasks where explicit reasoning is required. Neural networks, particularly DL models like CNNs [15,45], excel in handling the complexity of image data by recognizing intricate patterns and features [9]. Reinforcement learning, a cutting-edge approach, is valuable for optimizing sequential decision-making processes, such as directing autonomous drones for targeted data collection [46]. AI’s transformative impact extends to applications like land cover mapping, which involves the detailed identification and categorization of surface types across large areas. Change detection is another critical application, enabling the identification of temporal variations in the environment, such as urban sprawl or glacier retreat. Environmental monitoring benefits significantly from AI by providing precise, timely insights into phenomena like air quality, biodiversity loss, and the effects of climate change.

By automating feature extraction and analysis, AI significantly enhances the efficiency and accuracy of RS workflows. Traditional methods of analyzing satellite data were labor-intensive and prone to human error. However, AI can process massive datasets at unprecedented speeds, delivering high-resolution analyses in hours instead of weeks. This automation not only saves time but also enables near real-time monitoring and decision-making, which is essential for disaster response and resource management. For example, during a natural disaster, AI-driven systems can rapidly analyze satellite imagery to identify affected areas, assess damage, and guide rescue operations.

Moreover, AI facilitates integration with other emerging technologies, such as the Internet of Things (IoT) and geospatial information systems (GIS) [47]. This synergy enhances RS capabilities by enabling dynamic, multi-source data fusion, which provides a more comprehensive view of complex systems. In sum, AI is not just a tool for RS but a transformative force that reshapes the possibilities of what can be achieved in Earth observation and beyond [36].

#### 3.1.1. Case Study 1: AI-Driven Flood Mapping for Disaster Response

Floods are among the most devastating natural disasters, requiring rapid and precise assessments for timely intervention. AI-enhanced RS has been instrumental in flood monitoring and management. Recent studies have proven the effectiveness of DL models for automatic flood detection using Sentinel-1 SAR (Synthetic Aperture Radar) imagery. In a study by Ghosh et al., U-Net and Fully Convolutional Networks (FCNs) were applied to Sentinel-1 images for flood detection. The models achieved over 90% accuracy, outperforming traditional threshold-based approaches in identifying flood-affected areas. The integration of AI with SAR imagery is particularly valuable as radar data can penetrate cloud cover, making it ideal for monitoring extreme weather events. The European Space Agency (ESA) and other disaster response organizations have begun adopting similar AI-enhanced RS techniques for real-time flood assessment, improving emergency response and resource allocation [48].

#### 3.1.2. AI-Powered Wildfire Detection

In recent years, AI has been instrumental in enhancing wildfire detection capabilities. Advanced systems, such as AI-enhanced cameras and remote sensors, have been deployed to monitor vast forested areas. For instance, in California, early AI-driven detection tools have been utilized to identify wildfires promptly. These systems analyze data from various sources, including cameras and satellites, to identify anomalies indicative of fires. While these technologies have significantly improved situational awareness, challenges remain in preventing all wildfires, especially under severe conditions like strong winds and dry vegetation [49].

#### 3.1.3. AI in Regenerative Agriculture

AI and ML are transforming regenerative agriculture by improving soil carbon measurement and management practices. Traditional sampling methods are labor-intensive and often inconsistent. Companies like Agricarbon and Perennial are leveraging AI to automate and refine soil analysis. For example, Agricarbon uses automated soil core extraction vehicles, while Perennial employs digital soil mapping techniques. These AI-powered approaches enable more accurate predictions of soil carbon levels, reducing the need for frequent sampling and providing farmers with valuable insights to enhance productivity and sustainability [50].

#### 3.1.4. AI-Assisted Wildlife Monitoring

Conservationists are increasingly adopting AI to monitor endangered wildlife populations. In Costa Rica’s Osa Peninsula, researchers deployed 350 audio monitors to track the elusive Geoffrey’s spider monkeys. The recorded data, analyzed with the assistance of AI, revealed critical insights into the health of the wildlife refuge. This approach exemplifies how AI can handle large datasets and uncover ecological patterns, aiding biodiversity conservation efforts as numerous species face extinction risks [51].

### 3.2. ML Algorithms in RS

ML is the backbone of many RS applications, allowing for pattern recognition, classification, and predictive analytics.

#### 3.2.1. Support Vector Machines (SVM)

SVM is an effective supervised learning algorithm widely regarded for its effectiveness in solving classification tasks [52,53]. It is particularly well-suited for applications where the objective is to classify data into distinct groups [54]. For instance, in RS and geospatial analysis, SVM has been successfully employed to differentiate between urban areas and vegetation in satellite imagery, leveraging its robustness to handle high-dimensional datasets [55,56]. The strength of SVM lies in its ability to identify the optimal hyperplane that separates classes with the maximum margin. This characteristic is instrumental in ensuring high classification accuracy, even when the training data are sparse or imbalanced. By focusing on support vectors—the critical data points that influence the position of the hyperplane—SVM minimizes overfitting and augments generalization to unseen data [57].

Additionally, the versatility of SVM extends to handling non-linear relationships through kernel functions, such as the radial basis function (RBF) or polynomial kernels. These kernels project data into higher-dimensional spaces, making it feasible to separate classes that are not linearly separable in the original feature space [58]. This adaptability makes SVM a reliable choice in various fields, including bioinformatics, text classification, and image recognition [59]. Yang et al. explored the parallel processing approach to SVM. Building on the widely used hybrid parallel SVM framework, a novel hybrid parallel SVM method incorporating sample cross-combination was introduced [60].

The spectral and spatial characteristics of pervious and impervious surfaces pose significant challenges to image classification and information extraction in detailed, small-scale urban surface mapping. Recent advancements in classification techniques have centered on object-based approaches and segmentation methods. Salleh et al. determined the most effective classification method for extracting linear features by comparing pixel-based and object-based approaches, using WorldView-2 satellite imagery (Figure 2). Key linear features, such as roads, building edges, and road dividers, are the focus of this analysis [61].

Two algorithms—SVM and rule-based classification—were evaluated using different software platforms. The findings indicated that object-based classification outperformed the pixel-based approach, delivering higher resolution and more precise results. Rule-based classification exhibited strong performance, achieving an overall accuracy of 88.6% with ENVI (Version 5.3) and 92.2% with e-Cognition. Conversely, the SVM approach faced challenges, including misclassification of impervious surfaces and urban features, as well as mixed-object outcomes, resulting in an overall accuracy of 75.1%. Salleh et al. conducted a comprehensive comparison of the two classification methods, highlighting their respective strengths and limitations. It also offered valuable insights into the capabilities of the software tools employed, contributing to a deeper understanding of effective techniques for analyzing urban surface features [61].

#### 3.2.2. Random Forests (RF)

RFs are highly effective ensemble learning algorithms widely used in land-use classification, vegetation mapping, and other RS applications [62]. They operate by constructing a large number of decision trees during training and combining their outputs through majority voting (for classification) or averaging (for regression) [63]. This approach reduces the risk of overfitting and enhances generalization, making them well-suited for complex and high-dimensional datasets [64]. One of the key strengths of RFs is their ability to handle noisy, incomplete, or imbalanced data while maintaining accuracy. The algorithm is inherently robust to outliers and irrelevant features, as it relies on a random subset of features at each split, which prevents over-reliance on any single variable [65]. Additionally, RFs can provide feature importance rankings, enabling users to identify the most significant variables influencing the model’s predictions [66].

In RS, RFs are frequently used to classify land cover types, map vegetation distribution, and monitor environmental changes over time [67]. For example, they have been employed in analyzing satellite imagery to differentiate between urban and natural areas, assess forest health, and detect agricultural patterns [68]. Their efficiency in processing large-scale geospatial data makes them a preferred choice for applications requiring high computational performance and accuracy [69]. Moreover, RFs are versatile, requiring minimal parameter tuning and performing well across a wide range of tasks, from ecological modeling to medical diagnostics. Their ability to provide interpretable outputs, combined with their resilience to data variability, ensures their continued popularity in diverse scientific and practical fields [70].

Sun et al. proposed an enhanced RF method that improves classification accuracy and reduces inter-tree correlation [63]. The approach involved two key improvements: retaining high-performance classification and regression trees (CARTs) and minimizing their correlations. Each CART’s accuracy was evaluated on three reserved datasets, and the trees were ranked accordingly. A modified dot product method calculated the cosine similarity between CARTs, and a grid search identified an optimal correlation threshold. Trees with low accuracy and high correlation were removed, ensuring only the best-performing trees remain. Experimental results showed that the improved RF outperformed five baseline models, achieving higher classification accuracy and better G-means and out-of-bag (OOB) scores. Non-parametric tests confirmed significant differences between the proposed model and the others, demonstrating its superior performance. In conclusion, the proposed method enhanced RF efficiency by addressing key issues of accuracy and correlation, offering a robust framework for building more reliable ML models [63].

Soil moisture plays a crucial role in hydrologic and climate research, but high-resolution global datasets are scarce. Han et al. addressed this gap by using physics-informed ML to develop a high-resolution, long-term global surface soil moisture dataset (GSSM1 km). By integrating data from the International Soil Moisture Network (ISMN), RS [71,72], meteorological sources, and physical process models, they created the GSSM1 km dataset, which provides surface soil moisture (0–5 cm depth) at a 1 km spatial and daily temporal resolution, spanning from 2000 to 2020. Performance testing showed a root mean square error of 0.05 cm^3^/cm^3^ and a correlation coefficient of 0.9. Key factors influencing moisture levels included the Antecedent Precipitation Evaporation Index (APEI), evaporation, and longitude. This dataset is valuable for climate extreme studies and analyzing long-term trends.

Moreover, Islam et al. evaluated the use of RF regression (RFR), a ML model, as an alternative to the Soil and Water Assessment Tool (SWAT) for streamflow prediction in the Rio Grande Headwaters [14]. RS data were used for the RF analysis, with RStudio (Version 3.6.0) for processing. The RFR model outperformed SWAT in accuracy, especially with longer training periods (1991–2010, 1996–2010, 2001–2010), showing improved performance with extended training. Snow depth and minimum temperature were the top predictors. SWAT, while requiring more setup, showed acceptable performance in annual streamflow but needs improvements for monthly predictions. The data were split into training and validation sets using bootstrapping and OOB sampling. The train() function handled bootstrapping and cross-validation to reduce bias and overfitting. Hyperparameters (ntree, mtry, maxnodes) were optimized through grid or random search to improve performance on the validation set. A 10-fold cross-validation approach divided the data into 10 subsets, training the model on nine subsets and validating it on the remaining one. The caret package automatically selected the best hyperparameters based on the minimum RMSE. Ridge regularization was applied during training to prevent overfitting by adding a penalty term to the loss function.

The final trained model was evaluated on a separate validation set to assess its ability to predict flow data accurately. A flowchart of the RF model is illustrated in Figure 3. The study recommended refining SWAT’s snowmelt routine and exploring other hydrologic processes for better monthly flow estimates, contributing to improved streamflow prediction and water resource management in snowmelt-driven regions [14].

#### 3.2.3. K-Nearest Neighbors (KNN)

KNN is a straightforward, yet highly useful algorithm widely employed in the domains of image classification and clustering [73]. It operates on a simple premise: the class of a data point is determined based on the majority vote of its nearest neighbors in the feature space [74]. This non-parametric, lazy learning algorithm is particularly well-suited for applications in spectral image analysis, where the similarity between data points can be quantified effectively using distance metrics [75,76]. KNN’s strength lies in its simplicity and adaptability, as it requires minimal assumptions about the underlying data distribution. Despite its computational intensity for large datasets, optimizations such as k-d trees or ball trees can significantly enhance its performance, making it a versatile choice in scenarios requiring robust pattern recognition and grouping [77].

Forest fires significantly impact public health, ecosystems, and biodiversity while accelerating desertification and harming conservation areas. RS is a key method for identifying and monitoring fire-affected regions due to the similar spectral traits of burned zones. Pacheco et al. examined the capabilities of KNN and RF classifiers for identifying fire-damaged areas in central Portugal [78]. Data from Landsat-8, Sentinel-2, and Terra satellites were used alongside Jeffries–Matusita (JM) separability statistics to assess classification performance. The study analyzed a 93.40 km^2^ fire that occurred on 20 July 2019, across Santarém and Castelo Branco districts. Challenges such as spectral mixing, sensor resolution, and image registration timing were identified as contributors to classification errors, with commission errors ranging from 1% to 15.7% and omission errors between 8.8% and 20%. Classifier performance was assessed using receiver operating characteristic curves, with all areas under the curve values exceeding 0.88. Overall accuracy ranged between 89% and 93%, demonstrating the effectiveness of the KNN and RF algorithms for fire impact classification. These findings highlighted the potential of these methods for improving wildfire detection and environmental assessment.

Figure 4 illustrates the classification framework and methodology adopted in this study [78]. The KNN and RF algorithms were employed to classify fire-affected areas and evaluate the influence of various satellite imagery sources on their performance. The workflow for supervised classification of burned vegetation utilized multispectral data from Landsat 8/OLI, Sentinel-2/MSI, and Terra (ASTER/MODIS), with training samples derived through photointerpretation techniques. The process commenced with image composition, followed by the selection of training samples and spectral separability analysis to distinguish between burned and unburned areas. The KNN and RF algorithms were then applied to perform classification. Validation data, combined with confusion matrices, were used to assess classification accuracy and analyze the contributing parameters. This structured approach included key steps: preparing training datasets, conducting spectral separability analysis, applying classification algorithms, validating results, and performing accuracy assessments. By integrating diverse satellite data and rigorous validation techniques, the methodology provided a comprehensive evaluation of the classifiers’ effectiveness in detecting and mapping fire-affected areas [78].

#### 3.2.4. Clustering Algorithms (e.g., K-Means, DBSCAN)

Clustering algorithms, such as K-Means [79] and DBSCAN [80], play a vital role in analyzing RS data. These algorithms group similar data points into clusters based on their attributes, such as spectral, spatial, or temporal features [81,82]. RS data, often derived from satellite imagery, LiDAR, or hyperspectral sensors, is typically high-dimensional and contains diverse features. Clustering enables the extraction of meaningful patterns and facilitates land cover classification, anomaly detection, and other analytical tasks [83].

K-Means is one of the most widely used clustering algorithms in RS due to its simplicity and computational efficiency. It partitions the data into a pre-defined number of clusters (k) by minimizing the variance within each cluster. In RS applications, K-Means is often employed for unsupervised classification, where ground truth data are unavailable. For instance, K-Means can be used to segment satellite imagery into distinct land cover types such as forests, water bodies, and urban areas. However, its performance heavily depends on the initial selection of cluster centroids and the pre-set number of clusters, which may lead to suboptimal results in complex datasets. Additionally, K-Means assumes that clusters are spherical and equally sized, which may not always align with the structure of RS data [84].

DBSCAN, on the other hand, is a density-based clustering algorithm that excels in handling datasets with varying cluster shapes and sizes. It groups data points based on their density and identifies regions with a high concentration of points while marking sparse regions as noise. This makes DBSCAN particularly useful in RS for identifying irregular patterns, such as urban sprawl, forest fragmentation, or flood-affected areas. Unlike K-Means, DBSCAN does not require specifying the number of clusters beforehand, making it more adaptable to the diverse nature of RS data [85]. It also effectively handles noise, which is common in RS due to sensor errors or atmospheric interference. However, DBSCAN can struggle with datasets where the density varies significantly across clusters, as its performance depends on the selection of parameters such as the minimum number of points (minPts) and the radius (epsilon) used to define neighborhoods [86].

Both K-Means and DBSCAN have been extensively applied in RS to address a wide range of challenges. For instance, K-Means has been used for vegetation classification in hyperspectral imagery, urban land-use mapping, and glacier change detection. DBSCAN has proven effective in detecting oil spills in oceanic imagery, identifying clusters of deforestation in forested areas, and analyzing point clouds from LiDAR data [87]. Advanced RS tasks often combine these clustering techniques with other ML methods or domain knowledge to enhance accuracy and robustness [88].

Paddy rice is a cornerstone of Taiwan’s agricultural sector, making accurate identification of its area and distribution essential. Combining diverse image datasets with effective classification models is key to achieving this goal. However, different classification models applied to various image types often yield distinct outcomes, attracting significant attention from researchers. Wan et al. examined the classification of multispectral (WorldView-2) and hyperspectral (CASI-1500) imagery from agricultural lands in Yunlin County, Taiwan. Both unsupervised algorithms, such as K-means and PCA, and supervised methods, including LDA and DBSCAN, were utilized for paddy rice classification. All datasets, including ground truth information, are meticulously recorded and validated [89].

DBSCAN achieved approximately 5.4% higher accuracy than K-means while cutting computational time to just 15% of what LDA requires. When combined with PCA for feature selection, DBSCAN achieved around 90% classification accuracy while saving 95.1% of computational time compared to backpropagation neural networks (BPN). Despite its efficiency, DBSCAN’s accuracy was lower than that of supervised models like BPN. Nevertheless, its ability to minimize preprocessing and parameter tuning makes it a practical choice for geoscience applications, delivering fast, cost-effective classification [89].

The DBSCAN process involves defining Eps as the radius for determining neighboring points and specifying the minimum number of points (MinPts) required for a cluster. Each data point was tested to see if it lies within the Eps radius of a cluster center. If so, neighboring points are evaluated, forming a cluster group through a process called direct density reachable. The algorithm counts the points within the radius and checks if they meet the MinPts threshold. For instance, with MinPts set to 5 and Eps to 1, neighboring points within this range are identified. In cases where points form linear or rectangular patterns, the algorithm evaluates whether they meet the minimum density criteria, classifying them as density-reachable and suitable for clustering. This approach efficiently identifies clusters based on spatial density, even in irregular distributions (Figure 5).

#### 3.2.5. Gradient Boosting Machines (GBMs) and XGBoost

GBMs and XGBoost are powerful ML algorithms extensively applied in RS to tackle complex predictive and classification tasks [90,91]. These algorithms work by progressively merging weak models, usually decision trees, to build a strong predictive model. Their effectiveness in handling complex, noisy, and varied datasets makes them highly suitable for RS tasks, which frequently involve data from multiple sources like satellite imagery, hyperspectral data, and LiDAR [92].

GBMs are a type of ensemble method where decision trees are added sequentially to improve predictions by minimizing a chosen loss function [93]. Each new tree focuses on correcting the errors made by the previous trees in the ensemble. This process continues until a predefined stopping point, such as a set number of iterations or when the loss function reaches a certain level of convergence, is reached. In the context of RS, GBMs have proven effective for applications like land cover classification, mapping forest types, and forecasting crop yields [94]. Their ability to capture non-linear relationships and complex interactions between features is particularly advantageous when dealing with RS data that often exhibit intricate spatial and spectral dependencies. One of the strengths of GBMs is their flexibility in incorporating various types of data and their ability to handle missing values and outliers effectively. However, GBMs are computationally intensive and prone to overfitting, especially with noisy data. Careful tuning of hyperparameters such as the learning rate, maximum depth of trees, and the number of trees is critical to achieve optimal performance [95].

XGBoost, an advanced implementation of GBMs, addresses several limitations of traditional gradient boosting methods by optimizing both speed and accuracy. It achieves this through innovations such as regularization, which reduces overfitting, and tree pruning, which improves computational efficiency. Additionally, XGBoost supports parallel and distributed computing, making it highly scalable for large RS datasets. One more essential element of XGBoost is its ability to handle missing data during training by learning the best direction to split even with incomplete feature values [96].

In RS, XGBoost has been successfully applied to diverse tasks such as urban land-use and land cover (LULC) classification, forest biomass estimation, and the detection of environmental changes. For instance, in hyperspectral RS, XGBoost has been used to classify vegetation types by leveraging spectral signatures across hundreds of bands. Its robustness and interpretability make it a preferred choice for generating high-quality predictions while providing insights into feature importance, which can guide domain experts in understanding the underlying drivers of observed patterns [97]. The adaptability and high performance of GBMs and XGBoost have made them integral tools in RS workflows. They are often used in supervised classification tasks where labeled data are available, as well as in regression problems, such as predicting soil moisture or temperature from satellite observations. These algorithms have also been integrated with geospatial tools and cloud computing platforms to process massive datasets efficiently, enabling near-real-time decision-making for applications like disaster management and precision agriculture [98].

Although CNNs are fundamentally DL models, they can also be combined with ML classifiers in hybrid frameworks. For example, replacing the fully connected layers of CNNs with gradient boosting algorithms can significantly improve accuracy. Bui et al. investigated such an approach, integrating three advanced ML classifiers—XGBoost, LightGBM, and CatBoost with CNN-based feature extraction for land cover analysis in Hanoi, Vietnam [95]. In this review, CNNs are discussed in detail under DL models in Section 3.3 but are mentioned here solely in the context of their hybrid application with ML algorithms. The methodology involved (1) segmenting SPOT7 satellite imagery and extracting key features, including spectral data and spatial metrics, (2) normalizing attribute values and generating graph-based datasets, and (3) using these graphs as input for the models to classify six land cover types: House, Bare Land, Vegetation, Water, Impervious Surface, and Shadow. Results revealed that CNN-based XGBoost (overall accuracy = 0.8905), LightGBM (0.8956), and CatBoost (0.8956) outperformed traditional approaches. The combination of object-based image analysis and CNN-powered gradient boosting algorithms significantly enhanced classification accuracy, making it a powerful and reliable alternative for land cover mapping.

The CNN architecture proposed here draws inspiration from several successful models in land cover classification, incorporating essential elements like hidden layers, feature maps, activation functions, and loss functions. The hybrid model follows a systematic process, outlined below and illustrated in Figure 6. Step 1: High-resolution imagery is segmented using parameters such as scale, compactness, and shape, which were refined through an iterative process to achieve optimal boundaries for groups of similar pixels. After segmentation, spectral and spatial feature metrics were assigned to each object. Step 2: To ensure consistency across features with varying units and scales, normalization was applied. The normalized data were then visualized in two-dimensional space and used as input patches for CNN training. The segmented image objects were randomly divided into training, validation, and test datasets. Given the large size of the training data, a hold-out method was employed, representing the entire study area more effectively than cross-validation. Step 3: The CNN was trained using a categorical log-loss function, with fully connected layers serving as classifiers. Step 4: The training data were passed through the trained CNN again, extracting the final dense layer to generate a new training dataset, which was subsequently used to train three gradient boosting algorithms. This hybrid approach, combining CNNs with gradient boosting techniques, enhances land cover classification by harnessing the strengths of both methodologies.

Table 1 compares SVM, RFs, K-Nearest Neighbors (KNN), Clustering Algorithms (e.g., K-Means, DBSCAN), and GBMs including XGBoost, focusing on their characteristics, advantages, and disadvantages in the context of RS applications.

### 3.3. DL Algorithms in RS

DL algorithms have revolutionized RS by enabling the analysis of complex, high-dimensional data such as multispectral and hyperspectral images. Table 2 summarizes the characteristics, advantages, and disadvantages of the commonly used models from the perspective of RS. Key DL approaches include.

#### 3.3.1. Convolutional Neural Networks (CNNs)

CNNs are among the most prominent and widely adopted DL architectures in RS. They are specifically designed for structured image data, enabling automated extraction of spatial and spectral features from high-dimensional inputs [113]. While CNNs have occasionally been integrated into hybrid frameworks with ML algorithms (see Section 3.2.5), they remain a core class of DL models and are therefore discussed here in detail. As a specialized class of DL models, CNNs are designed to process structured data such as images, making them particularly well-suited for analyzing RS imagery acquired from satellites, drones, and aerial platforms. Their ability to automatically learn and hierarchically represent features from raw data have significantly advanced applications like land cover classification, object detection, and change detection [45].

CNNs derive their strength from a specialized architecture designed to process spatial information in images. Convolutional layers use learnable filters to extract features like edges, textures, and patterns in a structured hierarchy. Pooling layers help downsample the data, improving computational efficiency and reducing sensitivity to minor positional variations. Fully connected layers integrate these extracted features to facilitate tasks such as classification or regression. In RS, this architecture allows CNNs to effectively analyze intricate spatial patterns and spectral details commonly found in satellite imagery and other geospatial datasets [114].

CNNs have been widely adopted for a variety of RS applications. For instance, in land cover classification, CNNs have been used to differentiate between vegetation, water bodies, urban areas, and other land types by analyzing the spatial and spectral patterns in multispectral or hyperspectral imagery. In object detection tasks, CNNs excel at identifying features such as buildings, roads, and vehicles, which are critical for urban planning and disaster management. Additionally, CNNs have been applied to monitor environmental changes, such as deforestation or glacier retreat, by comparing temporal satellite images and detecting subtle differences. The ability of CNNs to integrate spectral information with spatial context has also proven invaluable in HSI. By capturing both the spectral signatures and spatial dependencies of pixels, CNNs can classify materials or vegetation types with remarkable accuracy [115]. Furthermore, the use of advanced architectures like FCNs and U-Nets has facilitated semantic segmentation tasks, enabling pixel-wise classification of RS images, which is essential for high-resolution mapping [10].

One of the key advantages of CNNs in RS is their ability to automate feature extraction, reducing the reliance on domain-specific expertise for manual feature engineering. This has accelerated the pace of innovation in applications such as precision agriculture, where CNNs are used to monitor crop health, and disaster response, where CNNs assist in damage assessment using post-disaster imagery [15]. Moreover, CNNs are scalable and can process large volumes of data, making them ideal for analyzing global satellite datasets [116].

Kattenborn et al. assessed a CNN-based segmentation method (U-Net) using training data obtained from visual interpretation of high-resolution RGB images captured by UAVs [114]. FCNs are effective at capturing contextual features over a wide receptive field (such as in orthoimages) while maintaining the spatial accuracy of these features, resulting in precise and detailed object segmentation (Figure 7). The approach was shown to reliably segment and map vegetation species and communities, achieving an accuracy of 84% or higher. A key takeaway was the method’s emphasis on spatial patterns rather than spectral data for plant identification at very high resolutions. This makes it compatible with low-cost, accessible UAV systems, extending its usability to a diverse range of users [114].

However, deploying CNNs in RS also presents challenges. The need for large, labeled datasets for training can be a bottleneck, as acquiring accurate ground truth data in RS is often costly and time-consuming. Additionally, the high dimensionality of RS data, particularly from hyperspectral sensors, can increase computational complexity and necessitate advanced hardware or cloud-based solutions. Despite these challenges, transfer learning and data augmentation techniques have been effectively used to mitigate data limitations by leveraging pre-trained models on related tasks [117].

As RS technologies continue to evolve, the integration of CNNs with advanced sensors and platforms will unlock new possibilities. Innovations such as 3D CNNs, which can process volumetric data, and hybrid models that combine CNNs with other ML techniques, are expected to further enhance the capabilities of RS applications [118]. Additionally, with the increasing availability of labeled datasets and advancements in computational resources, CNNs are poised to play an even more significant role in addressing global challenges such as climate change, urbanization, and disaster resilience.

#### 3.3.2. Recurrent Neural Networks (RNNs)

RNNs and their advanced forms, such as Long Short-Term Memory (LSTM) networks [119], are widely utilized for the time-series analysis of RS data. These models excel in processing sequential data by maintaining memory of past inputs, making them critical for tracking changes in vegetation, weather patterns, or other environmental parameters over time [120]. For example, LSTMs are highly effective in detecting subtle temporal trends in satellite imagery or sensor data, which is essential for monitoring phenomena such as deforestation, urban expansion, and seasonal crop health variations [121]. Furthermore, these models are increasingly employed in climate modeling and prediction tasks, where the ability to capture long-term dependencies in data are crucial for understanding complex systems like precipitation cycles, temperature fluctuations, and extreme weather events. The integration of RNNs and LSTMs with RS data also supports advancements in early warning systems, aiding in disaster risk reduction by providing accurate forecasts of floods, droughts, and other natural hazards [122,123].

Image classification has been a longstanding challenge in computer vision, with numerous models evolving from handcrafted feature-based methods to advanced DL techniques. Lakhal et al. focused on the classification of RS images, a task of critical importance to various real-world applications. The authors introduced a novel deep recurrent architecture that effectively integrates high-level feature descriptors to address this complex problem [124]. Experimental results highlighted that the proposed framework significantly outperformed previous methods across three widely recognized datasets, achieving a remarkable state-of-the-art (SOTA) accuracy of 97.29% on the UC Merced dataset.

Advances in DL have introduced cutting-edge methods for land cover classification. However, many existing DL-based approaches fail to account for the rich spatial relationships between land cover classes embedded in RS images. Tang et al. presented a novel deep relearning framework, referred to as Deep Relearning RNN (DRRNN), designed to address this gap [121]. Unlike traditional approaches, this innovative relearning strategy applied for the first time in DL-based land cover classification—demonstrated significant potential for enhancing classification accuracy. The proposed method introduced a class-correlated feature (CCF), extracted within a local window from an initial classification result, to capture both spatial autocorrelation and the spatial arrangement of land cover classes. By leveraging the sequential data-processing capabilities of RNNs, the CCF was transformed into a feature sequence, enabling the RNN to model dependencies between class labels. The classification process was refined iteratively through relearning, using the CCF alongside spectral–spatial features, until a predefined stopping condition was met.

The effectiveness of the DRRNN was validated on five RS datasets from diverse sensors and environmental conditions. Results showed that incorporating spatial autocorrelation reduced classification noise, while the inclusion of spatial arrangement corrects misclassified regions. Compared to SOTA DL methods, the proposed approach consistently achieved the highest accuracy, underscoring its value for land cover classification in RS applications. Figure 8 illustrates the extraction of the CCF. Misclassifications between buildings and roads often occurred due to similar spectral features. In residential areas, typical spatial patterns like “vegetation-shadow-building” make road-class pixels unlikely, and they were corrected in subsequent steps. Similarly, in road-dominated areas with high spatial autocorrelation, vehicles misclassified as buildings are reassigned to the road class. The CCF captured local spatial patterns, allowing the RNN to leverage spatial arrangement and autocorrelation in RS images, thereby improving classification accuracy.

#### 3.3.3. Generative Adversarial Networks (GANs)

GANs are applied in data augmentation and super-resolution tasks, such as enhancing the spatial resolution of satellite images. They are also used for creating synthetic imagery to train other models [125]. In recent years, advancements in RS technology have significantly expanded the application of RS images across various fields. However, the presence of clouds often obstructs these images, reducing their quality and usability. Consequently, removing clouds has become a critical preprocessing step in RS image analysis. While DL models have achieved remarkable results in image denoising, relatively little research has focused on applying deep neural networks (DNNs) to the challenge of cloud removal in RS images. To address this gap, Jin et al. introduced the Hybrid Attention Generative Adversarial Network (HyA-GAN), a novel model that combined the channel attention mechanism and the spatial attention mechanism within a generative adversarial network framework [126]. By integrating these attention mechanisms, HyA-GAN enabled the network to focus on critical regions of the image, enhancing its ability to recover data and produce high-quality cloud-free images. When evaluated against existing cloud removal models on the RICE dataset, HyA-GAN demonstrated superior performance, particularly in terms of peak signal-to-noise ratio (PSNR). These results underscored the model’s potential to effectively address the challenges associated with cloud removal in RS imagery [126].

Furthermore, Rui et al. introduced a method for synthesizing disaster RS images that represent multiple disaster types and varying levels of building damage using GANs, effectively mitigating the shortcomings of current datasets [127]. Conventional models faced challenges in translating multi-disaster images due to the varied nature of disasters and unintended alterations to non-building regions when processing entire images. To address these issues, two innovative models were introduced: the Disaster Translation GAN and the Damaged Building Generation GAN. The Disaster Translation GAN utilizes an attribute-based representation for disaster types and integrates a reconstruction mechanism to improve generator performance, enabling a single model to handle multiple disaster scenarios. Meanwhile, the Damaged Building Generation GAN employs a mask-guided strategy to selectively modify disaster-affected areas while preserving unaffected regions, ensuring more accurate image synthesis. Extensive qualitative and quantitative evaluations demonstrated the effectiveness of these approaches [127].

To evaluate the performance of the Damaged Building Generation GAN, the generated outputs were visualized. As illustrated in Figure 9, the first three rows show the pre-disaster images (Pre_image), post-disaster images (Post_image), and damaged building labels (Mask), respectively. The fourth row displays the generated images (Gen_image). The results clearly demonstrated that the model effectively modified the damaged areas while preserving regions unaffected by damage, such as intact buildings and the background. Moreover, the generated damaged structures were seamlessly integrated with the existing architectural features and their surroundings, resulting in highly realistic images. However, the synthetic damaged buildings lacked fine-textured details, highlighting an important aspect for future model refinement [127].

#### 3.3.4. Autoencoders

Autoencoders are a DL approach widely used in RS for tasks such as feature extraction, dimensionality reduction, and anomaly detection [128]. They comprise an encoder that transforms input data into a compact latent representation and a decoder that reconstructs the original input from this reduced form [129]. Training focuses on minimizing reconstruction error, allowing the model to learn meaningful and efficient data representations. In remote sensing, autoencoders are especially valuable for handling high-dimensional datasets, including satellite imagery and hyperspectral data. By reducing dimensionality, they facilitate data analysis while preserving essential information [130]. Autoencoders are also effective in detecting anomalies or changes over time, such as identifying deforestation or urban expansion, as well as isolating noise from the data [2,131,132,133]. Their ability to learn hierarchical features without manual feature engineering makes them especially valuable in automating RS workflows and improving accuracy in geospatial analysis and decision-making [134].

A novel DL-based approach using a Stacked Denoising Autoencoder (SDAE) was proposed to address the accuracy challenges in traditional RS image classification methods (Figure 10a). The model was constructed by stacking layers of Denoising Autoencoders, where each layer was trained sequentially using an unsupervised greedy layer-wise algorithm with noisy inputs to enhance robustness [133]. Features were further refined through supervised learning using a Back Propagation (BP) neural network, and the entire network was optimized via error backpropagation. Evaluated on Gaofen-1 (GF-1) satellite RS data, the method achieved total accuracy and kappa accuracy of 95.7% and 0.955, respectively—surpassing the performance of SVM and BP neural network (NN) classifiers.

This study classified each pixel of a RS image into a land cover category using an S × S image block centered on the target pixel as input to the SDAE. This minimized noise interference and utilized spectral, texture, and shape information without manual feature extraction. A 3 × 3 image block with 4-band gray values was used, resulting in an input vector of 3 × 3 × 4. Labels were vectors representing n categories, with the highest SDAE output value determining the classification. The process is displayed in Figure 10b. Figure 10c presents the classification results for the mountainous area. Notably, while SVM and BP misclassify several ARC pixels as SD, SDAE successfully identifies them correctly. These findings highlighted the method’s potential to significantly improve the accuracy of RS image classification [133].

To overcome the challenge of significant differences between two heterogeneous images, traditional unsupervised methods typically transform both images into a shared domain through various techniques such as alignment or transformation. However, these methods are computationally expensive and often struggle to balance the training process. In response, Shi et al. presented self-guided autoencoders (SGAE) for unsupervised change detection in heterogeneous RS images [2]. Unlike conventional approaches that seek to reduce the disparities between heterogeneous images to highlight changes, SGAE facilitated the flow of information within unlabeled data through self-guided iterations. Initially, an unsupervised network generated a basic change map, which was refined through filtering to produce reliable pseudo-labels. These pseudo-labeled samples were then fed into a supervised network to produce another change map. The resulting maps were combined to improve the reliability of the pseudo-labels, creating new pseudo-labeled samples for the self-guided network. This network was trained using both pseudo-labeled and unlabeled data. The iterative process enabled the network to continuously refine its feature extraction, improving its ability to classify. Experiments on four datasets, comparing this approach with several existing algorithms, demonstrated the method’s effectiveness and robustness, showing its potential to enhance unsupervised models by improving feature extraction and classification performance through a more adaptive learning framework [2].

#### 3.3.5. Transformer Models

Transformer-based architectures, originally designed for NLP, have rapidly gained prominence in RS tasks due to their remarkable ability to capture long-range dependencies and process sequential data effectively [135,136]. These models, such as BERT, GPT, and Vision Transformers (ViTs), are now being leveraged for complex tasks like scene classification and multi-label annotation in RS. Their attention mechanism allows for the efficient extraction of spatial and contextual features from satellite and aerial imagery, enabling them to identify intricate patterns across diverse landscapes. Unlike traditional CNNs, which operate on local patches of data, transformers can process the entire image in parallel, making them highly suitable for handling large-scale and high-dimensional remote-sensing data [135]. Additionally, their scalability and flexibility have allowed them to excel in multi-label annotation, where each image can be associated with multiple, overlapping classes, a challenge commonly encountered in geospatial analysis. The integration of transformers into RS workflows is transforming the field, offering significant improvements in the accuracy, speed, and interpretability of results.

Semantic segmentation of RS images focuses on categorizing each pixel into distinct, non-overlapping regions. This process is essential for various applications such as land cover classification, autonomous navigation, and scene analysis. Although DL methods have shown potential, there is still a lack of extensive research on effectively handling fine-grained details in RS images while coping with the high computational costs involved.

To tackle this issue, Boulila et al. introduced an innovative approach combining convolutional and transformer networks [137]. This design utilized convolutional layers with a small receptive field to capture fine-grained features, which were crucial for detecting small objects in high-resolution images. At the same time, transformer blocks were applied to capture larger contextual information. By merging convolutional layers for detailed local features and transformer self-attention for broader context, the method reduced the need for heavy downsampling, allowing the network to process full-resolution features directly. This approach proved particularly effective for detecting small objects. Additionally, it decreased the dependence on large datasets, which are often required by transformer-only models. Experimental outcomes confirmed the success of this approach in capturing both detailed and contextual information, achieving a mean dice score of 80.41%. This performance surpassed that of established models, including UNet (78.57%), Fully Connected Network (FCN) (74.57%), Pyramid Scene Parsing Network (PSP Net) (73.45%), and Convolutional Vision Transformer (CvT) (62.97%)—all tested under the same conditions and dataset [137].

**Table 2 sensors-25-05965-t002:** Characteristics, advantages and disadvantages of the DL algorithms presented in this section.

Model	Characteristics	Advantages	Disadvantages
CNNs	Specialized for grid-like data (e.g., images); uses convolutional layers to extract spatial features [138].	Excellent for spatial feature extraction; High accuracy for image classification and object detection tasks; Handles large image datasets well [139].	Requires large labeled datasets; Computationally intensive for high-resolution RS data; Limited capability in capturing temporal dependencies [140].
RNNs	Designed for sequential data; uses feedback loops for temporal relationships [141].	Ideal for processing time-series data (e.g., vegetation growth or land cover changes); Captures temporal dependencies [142].	Prone to vanishing/exploding gradient issues; Struggles with long-term dependencies in large datasets; Inefficient for spatial data analysis [143].
GANs	Comprises a generator and discriminator network; used for generating synthetic data [144].	Effective in generating realistic synthetic RS images; Useful for data augmentation in underrepresented areas [145].	Training is unstable and requires careful tuning; Can remove noise; High computational cost; Vulnerable to mode collapse, where the generator produces limited variety [146].
Autoencoders	NN trained to encode input data into a compressed representation and decode it back [147].	Effective for dimensionality reduction and feature extraction; Suitable for anomaly detection in RS (e.g., wildfire monitoring) [148].	Struggles with generating high-quality reconstructions for complex data; Requires labeled data for specific tasks; Limited ability to capture temporal features [129].
Transformer Models	Utilizes self-attention mechanisms; excels at capturing long-range dependencies in data [149].	Exceptional performance for processing multi-modal RS data (e.g., combining spatial and temporal data); Handles large-scale datasets [150].	Computationally intensive, especially for high-resolution imagery; Requires significant memory resources; Needs extensive training data [151].

To synthesize the methodological advances discussed in this section, Figure 11 presents a conceptual workflow of RS integrated with AI, ML, and DL. The diagram outlines the progression from data acquisition and preprocessing to AI/ML/DL algorithms, and illustrates how outputs translate into diverse application areas. A feedback loop emphasizes the ongoing role of human validation, domain expertise, and XAI in refining and deploying these models effectively.

To ensure the review is not only descriptive but also evaluative, we include a critical comparison of methods, highlighting their relative performance, limitations, and suitability for different RS tasks. While the preceding table summarizes methodological characteristics, advantages, and disadvantages, readers may also benefit from quantitative evidence of comparative effectiveness. Table 3 compiles reported performance metrics from representative studies, offering a more evidence-based perspective on the strengths and weaknesses of different approaches in RS applications.

As shown in Table 3, performance varies substantially across methods and tasks. Classical ML algorithms such as SVM and RF remain effective in specific contexts: RF and KNN, for example, achieved accuracies above 89% in wildfire damage mapping [78], while rule-based classifiers outperformed SVM in detailed urban surface extraction [61]. However, their limitations become evident with complex or mixed-feature datasets. Deep learning methods generally provide higher accuracies, with CNN-based U-Net models reaching ≥84% for vegetation mapping [114] and recurrent frameworks like DRRNN setting state-of-the-art benchmarks for land cover classification [121]. At the same time, DL approaches require extensive training data, high computational resources, and careful tuning, which can hinder their practical deployment. Generative models such as GANs show promise in enhancing data quality (e.g., cloud removal [126]), but remain unstable and less mature compared to supervised classifiers. Overall, while traditional ML methods continue to offer robust, resource-efficient solutions in constrained settings, DL architecture demonstrates superior accuracy and generalization, particularly for high-resolution imagery and complex RS tasks.

## 4. Challenges and Prospects of AI in RS

While Section 3 surveyed methodological advances and applications of AI, ML, and DL in RS, this section shifts focus to the limitations and open challenges identified across the literature. The purpose here is not to introduce new methods, but to synthesize recurring difficulties that constrain progress such as data heterogeneity, computational demands, model interpretability, and transferability. By separating advances (Section 3) from challenges (Section 4), we aim to provide readers with both an overview of the state-of-the-art and a critical discussion of the obstacles that remain.

### 4.1. Challenges

Despite the remarkable advancements in AI-driven RS, several key obstacles hinder its full potential. These challenges range from data quality limitations to computational demands, impacting the scalability and reliability of AI applications in geospatial analysis.

#### 4.1.1. Data Quality and Diversity

RS data frequently encounter challenges such as noise, incomplete information, and inconsistent resolutions, stemming from the diverse array of sensors and platforms used for data collection [152]. These issues can significantly impact the accuracy and reliability of downstream analyses [153]. Advanced AI models, particularly DL algorithms, rely heavily on high-quality, well-annotated datasets for effective training, validation, and testing [7]. However, the scarcity of standardized, annotated datasets poses a significant obstacle, impeding the development and benchmarking of robust and transferable AI models. Furthermore, the inherent heterogeneity of RS data sources—including variations in spatial, spectral, and temporal characteristics—compounds the difficulty of achieving model generalization. Addressing these challenges requires concerted efforts to improve data quality, establish comprehensive annotation protocols, and create benchmark datasets that accommodate the diverse nature of RS applications [154].

#### 4.1.2. Computational Requirements

Training advanced AI models demands substantial computational power, memory, and storage resources. These requirements can be a limiting factor for researchers and organizations with limited access to high-performance computing (HPC) infrastructure [155]. Moreover, deploying such models on edge devices, like drones or in situ sensors, remains a significant challenge due to the trade-offs between model complexity and real-time processing capabilities. Bortnyk et al. examined the obstacles in AI training, emphasizing data constraints, high computational expenses, and the necessity for resilient models. They explored cutting-edge solutions such as synthetic data creation, optimized neural architectures, and techniques for enhancing robustness while underscoring the significance of AI model interpretability [156].

#### 4.1.3. Model Interpretability and Explainability

AI models, particularly DL networks, are often treated as “black boxes” due to their lack of transparency in decision-making processes [157]. This opacity poses a challenge for critical applications like disaster management or defense, where understanding the rationale behind predictions is as important as accuracy [158]. Without clear explanations, stakeholders may struggle to trust or act upon AI-generated insights, especially in high-stakes scenarios where decisions impact human lives and infrastructure. To address this issue, researchers have developed various techniques for model interpretability and explainability, such as feature attribution methods (e.g., SHAP, LIME), attention mechanisms, and surrogate models that approximate complex architectures with more understandable representations [159]. Regulatory frameworks like the EU’s General Data Protection Regulation (GDPR) further emphasize the need for explainability, particularly in AI systems that affect individuals’ rights and freedoms. In addition, ethical and societal concerns deserve greater attention. One major issue is algorithmic bias, which can arise when training datasets are geographically or socioeconomically unbalanced, leading to uneven performance and reinforcing existing inequities. Another concern is privacy, since high-resolution RS imagery combined with AI analysis can inadvertently reveal sensitive information about individuals or communities. Finally, the societal impact of AI-driven RS systems extends to areas such as disaster management, environmental monitoring, and urban planning. While these technologies offer clear benefits, they may also shift power dynamics among governments, corporations, and local communities. Ensuring transparency, accountability, and stakeholder participation is therefore essential to maximize societal benefits and minimize risks.

Improving model interpretability not only enhances transparency but also aids in identifying biases, debugging errors, and refining model performance. In domains such as healthcare, finance, and autonomous systems, interpretable AI ensures ethical and accountable deployment, fostering greater adoption and trust among users. As AI continues to evolve, the balance between model complexity and interpretability remains a key challenge, driving ongoing research into more explainable DL architectures. Dobson et al. examined various strategies that humanist scholars and ML critics can use to analyze and interpret Transformer-based DL models. By revealing different aspects of what Transformers “learn” during training and language processing, these approaches help illuminate their inner workings. Moreover, they are especially valuable for digital humanists who engage with these models through the lens of tool criticism, treating computational tools as essential components of interpretive research. The need for explainable AI (XAI) in RS is increasingly recognized but remains underdeveloped [160].

#### 4.1.4. Transferability and Domain Adaptation

AI models trained in one region or dataset may not perform well when applied to other regions or datasets due to differences in environmental, atmospheric, and sensor conditions. Domain adaptation techniques, which aim to improve model transferability across domains, are still in their infancy in the context of RS [161]. Multi-Modal Unsupervised Image-to-Image Translation (MUNIT) enhances domain adaptation by incorporating multiple approaches for unsupervised image-to-image translation [162,163]. By leveraging disentangled representations, MUNIT effectively separates content and style information, enabling the generation of diverse and realistic translated images across different domains [164]. This advancement increases its versatility, allowing it to handle a wide range of adaptation tasks with greater flexibility and efficiency [165].

Furthermore, MUNIT’s architecture is designed to support multimodal outputs, meaning it can generate multiple plausible translations for a given input. This characteristic is particularly beneficial in scenarios where variability is essential, such as medical imaging, autonomous driving, and artistic style transfer. By learning domain-invariant content representations while maintaining domain-specific style attributes, MUNIT facilitates more robust and adaptable model performance across different datasets. Additionally, MUNIT employs a combination of adversarial training and reconstruction losses to ensure high-quality image generation while preserving essential structural details. This approach not only improves image realism but also enhances model generalization, making it highly effective for real-world applications. Figure 12 illustrates the mechanism of a typical DL-based domain adaptable model, highlighting the key components involved in unsupervised image-to-image translation and their role in achieving effective domain adaptation.

### 4.2. Prospects

The future of AI in RS is shaped largely by methodological innovations such as multimodal learning, model efficiency, explainability, and collaborative frameworks. At the same time, certain application domains, most notably climate and environmental monitoring represent critical frontiers where these methodological advances will have profound impact. To reflect this, Section 4.2 discusses methodological prospects first, followed by a distinct application-driven prospect.

#### 4.2.1. Advances in Data Fusion and Multimodal Learning

AI is poised to excel in integrating data from multiple sensors and platforms, such as optical, radar, and LiDAR, to generate comprehensive insights. Multimodal learning approaches can enhance the understanding of complex environmental phenomena by leveraging the complementary strengths of various data types [166,167,168]. DL models are essential for earth observation, leveraging vast RS data. Multimodal learning enables the fusion of diverse data sources, especially when combining images with tabular data or varying temporal and spatial resolutions. However, as modality differences increase, fusion complexity rises, making DL models less transparent and explainable—critical for sensitive applications [169]. Gunther et al. explored how the RS community addresses these challenges, drawing insights from other fields and also highlighted promising explainability methods for multimodal deep networks to enhance their effectiveness in RS applications [170].

Multimodal contrastive learning aligns text with corresponding images as positive pairs. Traditional approaches rely on fixed sentence structures for text and global features for images, with some methods incorporating local features. While effective, this approach enforces a strict identity constraint, which may not fully capture the complexity of RS images. RS images present unique challenges. They contain rich ancillary information beyond object features and are difficult to describe in a single sentence. To address these challenges, Zhang et al. proposed a multimodal contrastive learning method for RS image feature extraction, introducing a tripartite relaxation strategy to enhance flexibility in three key areas [166].

First, constraints on text and image inputs were relaxed by incorporating learnable parameters in both the language and image branches. Instead of relying on predefined sentence structures and fixed image features, this enabled adaptive textual descriptions and richer feature extraction. Second, multimodal alignment was improved by linking semantic information to specific image regions. Unlike unimodal approaches that lack semantic constraints when selecting image patches, this method ensured a more meaningful correspondence between textual semantics and local image features. The effectiveness of this approach was validated on four datasets. On the PatternNet dataset, it achieved 91.1% accuracy with just one-shot learning, demonstrating its capability in RS image feature extraction [166].

#### 4.2.2. Improved Model Efficiency

Research on lightweight AI architectures and model compression techniques is making significant strides, enabling the deployment of AI models on resource-constrained devices like drones and edge sensors [171,172]. These advancements will enhance real-time processing capabilities, broadening AI’s applicability in RS [173]. Dantas et al. examined model compression techniques in ML to enhance efficiency in resource-limited environments like mobile devices, edge computing, and IoT systems. As ML models grow more complex, their computational demands increase, making compression essential for maintaining performance while reducing resource use [171]. Dantas also highlighted hybrid methods that combine multiple compression techniques and adaptive frameworks for optimal strategy selection. Practical examples demonstrate their impact in real-world applications. By balancing model complexity and efficiency, compression ensures scalable and sustainable AI advancements [171].

RS images present challenges for object detection due to their intricate backgrounds and numerous small targets. To improve detection performance, Gong et al. introduced YOLO-AEW, an optimized adaptation of YOLOv8s [174]. A lightweight Adaptive-weight Subsampling (ACS) module was integrated to minimize the model’s parameter count while maintaining effective feature extraction. Additionally, an Efficient Multi-scale Attention (EMA) module enhanced feature representation and detection accuracy. To further refine training, improve stability, and enhance robustness, the Wise-IoU bounding box loss function was employed. Comprehensive experiments on the High-Resolution RS Dataset (HRRSD) validated the effectiveness of YOLO-AEW. Results showed a 1.1% increase in mAP alongside an 11.3% reduction in parameters compared to the baseline, demonstrating improved detection accuracy and reliability [174].

Moreover, detecting small targets in RS images is inherently challenging due to issues such as noise and the limited availability of detailed information about these targets. To address these challenges, Elhamied et al. introduced an improved model designed to boost detection performance for small targets [175]. The model started with adjustments to the feature extraction process by integrating an attention mechanism, which enhances the quality of the extracted features. A transformer block was added to improve the representation of the feature map, and a bidirectional feature pyramid network with attention guidance further refined the extraction of discriminative information. This was achieved through a dynamic, sparse attention mechanism that selectively focused on key features from earlier layers. Additionally, top–down pathways were employed to ensure better feature integration throughout the network. To improve the alignment between detected and ground-truth bounding boxes, the model introduced a Rectified Intersection Over Union loss function, which maintained better consistency in shapes. Extensive testing on the DIOR, VHR-10, and VisDrone2019 datasets demonstrated the model’s effectiveness. The results showed significant improvements in mAP for small targets, overall mAP, model efficiency, and Frames Per Second (FPS). A comparison with the original YOLOv8s architecture revealed clear advancements in detection accuracy, underscoring the effectiveness of these enhancements for small-target detection in RS images [175].

#### 4.2.3. Development of Explainable AI (XAI)

The growing focus on explainability is leading to the development of models that provide interpretable insights, thereby increasing trust in AI applications. Techniques such as saliency maps, attention mechanisms, and rule-based systems are being integrated to make AI predictions more transparent and actionable [176]. Ishikawa et al. introduced “What I Know (WIK),” a XAI method that verified the reliability of DL models by showing training data examples similar to the input being processed. Demonstrated in RS image classification, WIK helped validate model inferences by ensuring the training dataset was appropriate for the task [176]. By checking if selected examples resemble the input data, the model was trained on similar feature distributions. This method considered both data similarity and task relevance. Successfully tested on a Sentinel-2 image dataset, WIK can be applied to various ML tasks, including classification and regression.

A comparison of results across various similarity metrics, as established in Figure 13, revealed that tL^2^-norm provides the most consistent outcomes, particularly for the river and industrial images. In the case of river images, the widths and curvatures closely matched those of the target image in both the -norm and maximum norm results. For industrial images, the sizes, shapes, and colors of the buildings aligned more closely with the target data using the L^2^-norm than with other metrics. While 3 out of 6 images showed similar results with both the L^2^-norm and maximum norm, the spatial scales of the structures in the residential and industrial images were better matched to the target data by the L^2^-norm. Based on these observations, the L^2^-norm was selected as the preferred similarity measurement method for WIK in the subsequent discussion.

The result from WIK allowed for an investigation into the data used to train the model and the instance pairs the model identified as similar. XAI methods aimed to explain model inferences to humans, and it is crucial to consider how these results are presented. If the presented pair appears reasonably similar, it suggested that the model extracts feature in a manner consistent with human intuition. This helped verify whether the model’s abstraction was both logical and aligned with human understanding. Additionally, it provided an opportunity to assess whether the training dataset sufficiently covered the feature space needed to accurately estimate the target’s positioning. If the pair does not seem similar, it may indicate that the training dataset failed to adequately cover the feature space surrounding the target or that the model was undertrained. In such cases, the inference result was generally deemed unreliable [176].

#### 4.2.4. Incorporation of Unsupervised and Semi-Supervised Learning

To address the scarcity of labeled datasets, unsupervised and semi-supervised learning approaches are being explored [177]. These methods leverage unlabeled data for feature extraction and representation learning, reducing the dependency on manual annotation. Li et al. introduced a semi-supervised approach for RS image scene classification, leveraging prototype-based consistency and a large set of unlabeled images [178]. The method first employed a feature enhancement module to extract more discriminative features by emphasizing foreground regions. Then, a prototype-based classifier was integrated to ensure consistent feature representations. Extensive experiments on the NWPU-RESISC45 and Aerial Image Dataset (AID) demonstrated the effectiveness of the proposed approach, achieving accuracy improvements from 92.03% to 93.08% on NWPU-RESISC45 and from 94.25% to 95.24% on AID, surpassing existing SOTA methods [178].

The effectiveness of a classification model improves with accurately labeled samples, which are often scarce in RS imagery. To address this challenge, semi-supervised learning techniques leverage both labeled and unlabeled data to enhance classification accuracy. Kothari et al. introduced a self-learning semi-supervised classification model based on a NN as the core classifier [179]. To overcome the limitations of conventional methods, the model incorporates an efficient neighborhood information learning approach. Identifying the most relevant neighborhood information for unlabeled samples was both crucial and challenging. To tackle this, two methods for generating similarity matrices were proposed that refine NN learning. The first method captured mutual neighborhood information, while the second utilized the class-map of unlabeled samples. A classifier trained on labeled data predicted the class labels of unlabeled samples, and the collaborative neighborhood information derived from both matrices further enhanced the proposed model. Experimental evaluations on three multispectral and one hyperspectral RS dataset demonstrated its superiority over existing SOTA methods. Performance was assessed using multiple metrics, including overall accuracy, kappa coefficient, precision, recall, dispersion score, and the Davies–Bouldin (DB) index [179].

#### 4.2.5. Collaborative Frameworks and Open Data Initiatives

Collaborative efforts, including open-source AI platforms and publicly accessible RS datasets, are driving innovation and broadening access to valuable resources. These initiatives allow researchers globally to participate in the development and refinement of AI models [7]. The efficient processing, analysis, and interpretation of large RS datasets using advanced technologies on RS cloud computing platforms (RS-CCPs) has become increasingly crucial [180]. However, most current RS-CCPs are mainly designed to enhance data storage and computing performance for general visual tasks, often neglecting important RS features like large image sizes, significant scale differences, multiple data channels, and the integration of geographic knowledge. These shortcomings can negatively impact both the efficiency and accuracy of computations.

To address these issues, Zhang et al. developed LuoJiaAI, a platform that combined an extensive sample database (LuoJiaSET) with a specialized DL framework (LuoJiaNET) [181]. This platform achieved exceptional results in five key RS interpretation tasks: scene classification, object detection, land-use categorization, change detection, and 3D reconstruction from multiple viewpoints. More detailed information on its practical applications is available in the LuoJiaAI industrial application report. In addition, LuoJiaAI was an open-source remote sensing-based cloud computing platform that complied with the latest Open Geospatial Consortium (OGC) standards, facilitating the creation and sharing of Earth-related AI models and products with standardized datasets. By effectively combining databases and DL frameworks through an intuitive collaboration system, LuoJiaAI showcased its significant potential for advancing highly accurate RS mapping applications [181].

#### 4.2.6. Application-Driven Prospect: Climate and Environmental Monitoring

Climate change has become a major driver of natural hazards and extreme weather events, resulting in profound environmental, social, and economic impacts [182]. Rising rainfall, sea levels, and shifting climate patterns have amplified the likelihood of flooding and other catastrophic events. AI-powered real-time disaster response systems, which utilize climate data from meteorological stations, satellite imagery, and sensor networks, can help predict and identify conditions that lead to disasters such as hurricanes, floods, wildfires, and volcanic eruptions. Beyond weather-related effects, climate change is also threatening biodiversity, altering habitats, genetic diversity, and species distribution. AI-driven monitoring systems, using satellite imagery and sensor data, can help detect and combat threats to biodiversity, such as illegal logging and poaching [183].

AI is playing an increasingly important role in environmental conservation. For instance, researchers are utilizing AI-powered acoustic monitoring in combination with drones, surveillance cameras, and satellite imagery to track endangered wildlife [184,185]. Vegetation, including forests and other plant cover, is essential for maintaining biodiversity by supporting oxygen production, enhancing water filtration, preventing land degradation, and stabilizing the water cycle [186]. A study by Akter et al. examined how AI-driven climate innovations impact industries known for high resource consumption and pollution. The research introduced and validated an AI model that categorizes key factors into three main dimensions and nine subcategories [187]. Focusing on the fast fashion industry, the findings indicated that AI-driven climate technologies positively affect both ecological and market outcomes, with environmental performance acting as a bridge between the two. The study underscored the importance of AI in developing strategies for climate resilience, adaptation, and mitigation.

RS techniques, such as DNN for vegetation mapping, are increasingly used to monitor and protect ecosystems, ensuring ecological balance and the preservation of biodiversity [186]. These AI applications provide innovative solutions to the many climate-related challenges we face, contributing to both environmental protection and resilience. Additional examples of AI-driven climate solutions can be found in Table 4. Unlike the preceding subsections, which emphasize methodological and structural advances, this subsection highlights an application-driven prospect, underscoring climate and environmental monitoring as a priority area where AI–RS integration can deliver transformative societal benefits.

## 5. Conclusions

The convergence of RS and AI has transformed the way we analyze and interpret Earth observation data, leading to unprecedented advancements in environmental monitoring, disaster management, urban planning, and agriculture. AI-driven techniques, particularly ML and DL, have significantly improved the accuracy, efficiency, and automation of RS applications. Traditional RS methods, which relied on manual or semi-automated techniques, often struggled with the increasing volume, complexity, and heterogeneity of data. However, AI has revolutionized the field by enabling sophisticated feature extraction, classification, object detection, and predictive modeling, reducing human intervention while enhancing the reliability of results. Models such as CNNs, RNNs, GANs, and transformer-based architecture have demonstrated remarkable success in handling high-dimensional, multimodal, and time-series data.

Despite these advancements, challenges remain that must be addressed for the broader adoption and effectiveness of AI in RS. Issues related to data quality, model generalization, explainability, and computational requirements continue to pose significant barriers. AI models trained on specific datasets often struggle with transferability across different geographical regions, atmospheric conditions, and sensor types, necessitating the development of robust domain adaptation techniques. Additionally, the black-box nature of many DL models raises concerns regarding interpretability and trust, especially in critical applications such as disaster response and environmental policymaking. The integration of XAI techniques, such as attention mechanisms, saliency maps, and rule-based approaches, is crucial for making AI-driven RS models more transparent and trustworthy.

Furthermore, the increasing reliance on AI introduces ethical considerations, including biases in training datasets, privacy concerns related to high-resolution satellite imagery, and the potential misuse of AI-generated insights. Addressing these challenges requires interdisciplinary collaboration between RS scientists, AI researchers, policymakers, and domain experts. The development of standardized datasets, open-source AI frameworks, and collaborative research initiatives can facilitate knowledge sharing and innovation in the field. Additionally, advances in edge computing, cloud-based AI processing, and multimodal data fusion will enable real-time and scalable AI applications in RS, further expanding its impact.

Looking ahead, future research should focus on enhancing the efficiency, interpretability, and adaptability of AI models for RS applications. Hybrid AI approaches that combine traditional physical models with data-driven learning techniques could improve model robustness and generalization. The integration of AI with emerging technologies such as the IoT, GIS, and HPC will further enhance RS capabilities. As AI continues to evolve, its role in global-scale environmental monitoring, climate change mitigation, and sustainable development will become increasingly vital. By addressing current challenges and leveraging future technological advancements, AI-driven RS will play a critical role in shaping data-driven decision-making for a more resilient and sustainable planet.

## Figures and Tables

**Figure 1 sensors-25-05965-f001:**
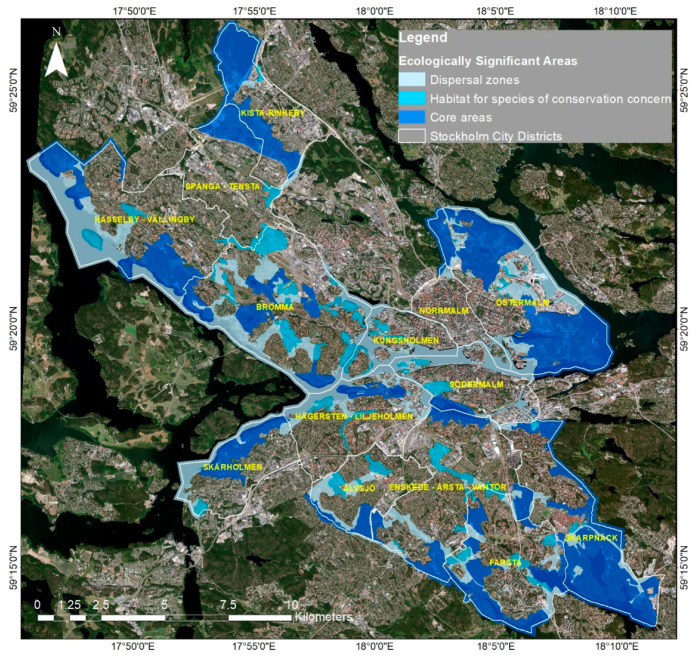
The study area is outlined by district boundaries in white, labeled in yellow. Ecologically significant green infrastructure areas are shown in varying shades of blue, with WorldView-2 imagery from 4 July 2018, as the backdrop [39].

**Figure 2 sensors-25-05965-f002:**
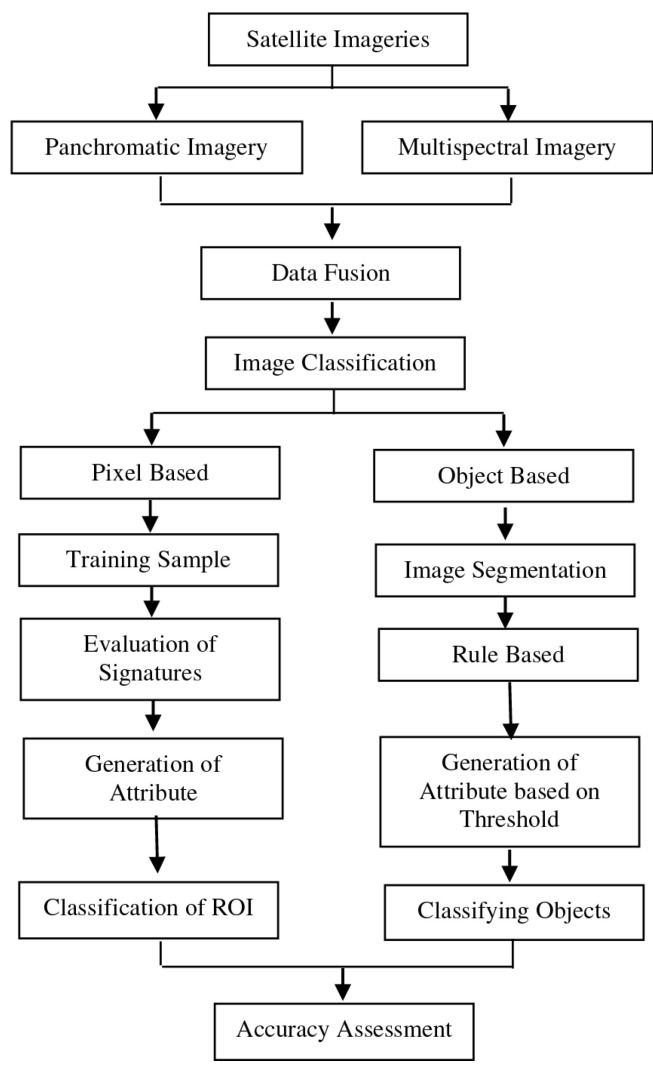
Methodological framework adapted from [61]. The framework illustrates the comparison of pixel-based and object-based classification methods for extracting linear features.

**Figure 3 sensors-25-05965-f003:**
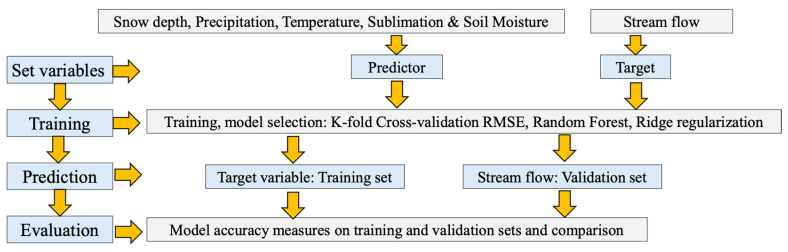
Flowchart illustrating the relationship between the predictors and target variable in the RF model. Inspired by [14].

**Figure 4 sensors-25-05965-f004:**
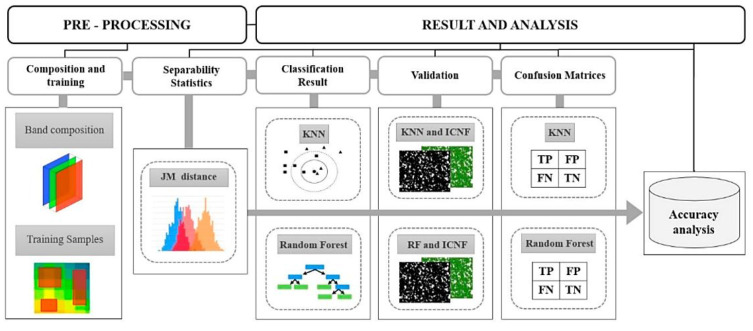
Flowchart of the methodology adapted from [78]. The diagram illustrates the sequential steps involved in the proposed approach, including data acquisition, preprocessing, feature extraction, model training, and performance evaluation. This visual representation highlights the logical workflow of the methodology and provides readers with a clearer understanding of how each component contributes to the overall process.

**Figure 5 sensors-25-05965-f005:**
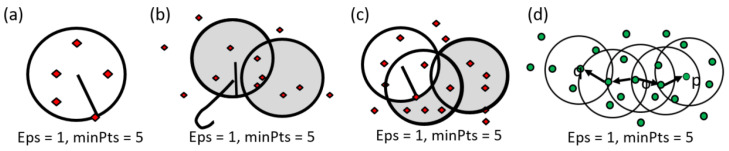
Visualizing the steps of DBSCAN: (**a**) Parameter definition; (**b**) Direct density reachability; (**c**) Indirect density linkage; and (**d**) Density connectivity [89].

**Figure 6 sensors-25-05965-f006:**
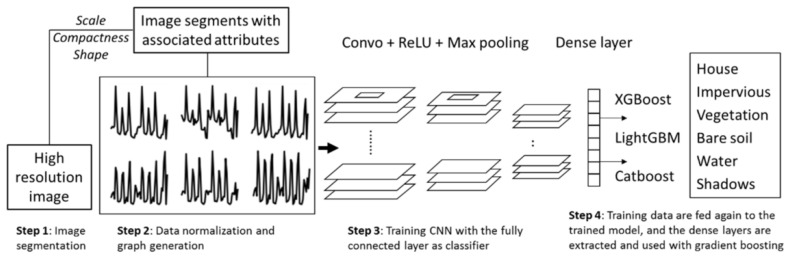
Object-Oriented CNN Integrated with Gradient Boosting Techniques [95].

**Figure 7 sensors-25-05965-f007:**
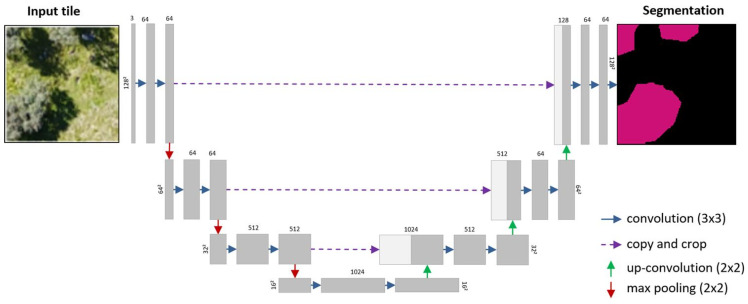
A diagram depicting CNN-based segmentation using the U-Net architecture. The input tile (**left**), a 128 × 128 pixel section, undergoes multiple convolutional layers (encoding layers) that extract spatial patterns at different scales (gray). Max pooling is then used to downsample the feature maps, reducing resolution before further convolutions refine the representation. In the final phase, decoding layers (up-convolutions) reconstruct spatial features from multiple scales to delineate the target class (**right**, pink) [114].

**Figure 8 sensors-25-05965-f008:**
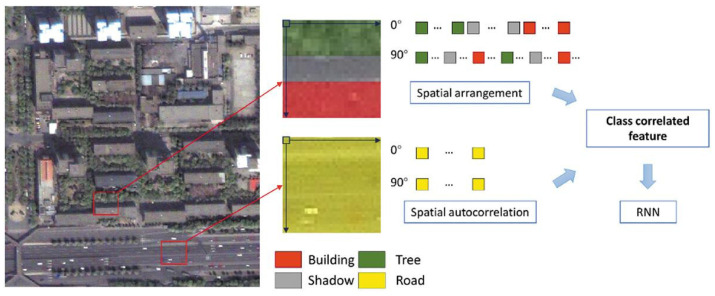
The extraction process of the CCF, incorporating spatial arrangement and autocorrelation features [121].

**Figure 9 sensors-25-05965-f009:**
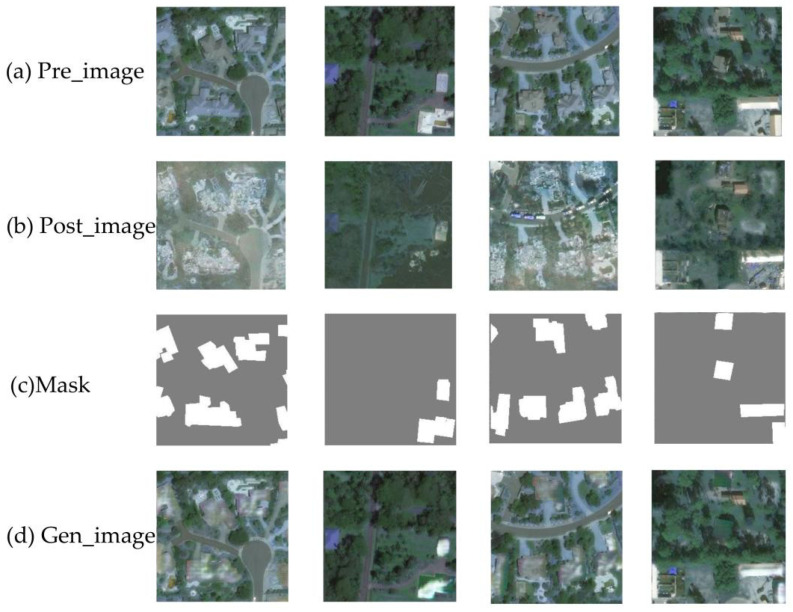
Results of damaged building generation: (**a**–**d**) correspond to the pre-disaster images, post-disaster images, masks, and generated images, respectively. Each column displays a pair of images, with four sample pairs provided [127].

**Figure 10 sensors-25-05965-f010:**
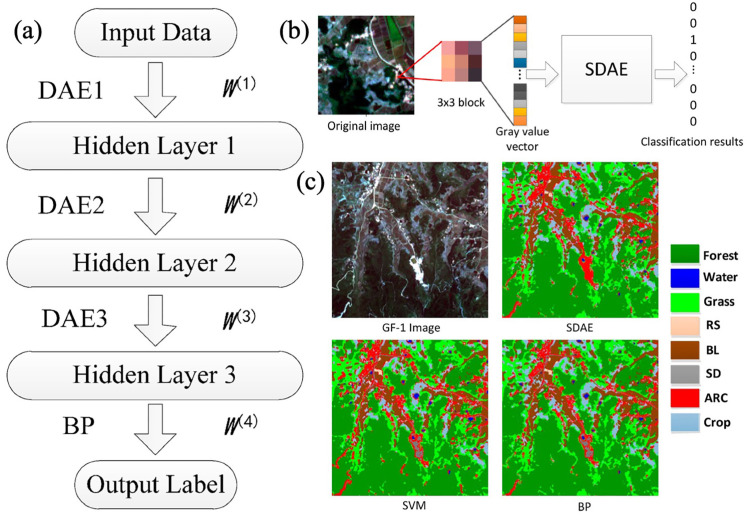
(**a**) SDAE, (**b**) Process of the RS image classification method using SDAE, (**c**) Classification outcomes for the mountainous area using different methods [133].

**Figure 11 sensors-25-05965-f011:**
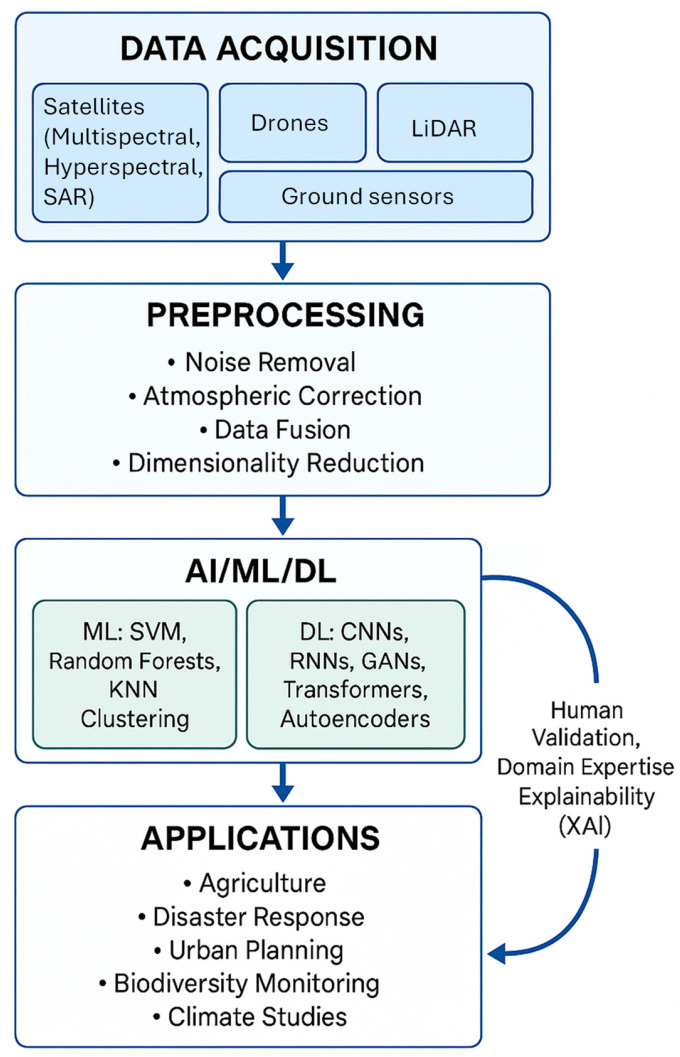
Workflow of RS integrated with AI/ML/DL, showing data acquisition, preprocessing, algorithms, applications, and a feedback loop for human validation and XAI.

**Figure 12 sensors-25-05965-f012:**
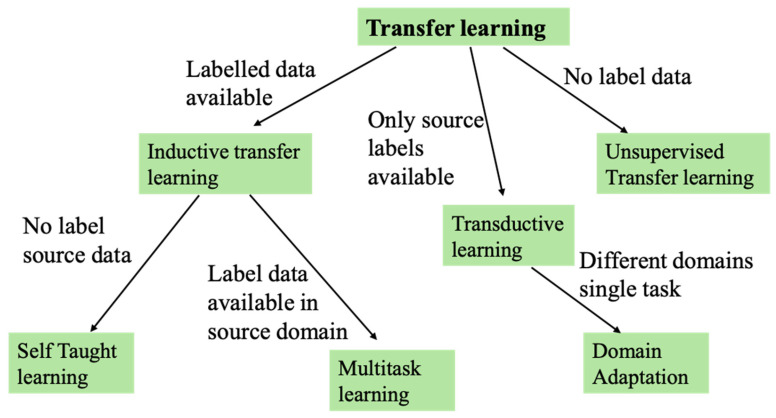
Mechanism of a DL-based domain adaptable model. Adopted from [162].

**Figure 13 sensors-25-05965-f013:**
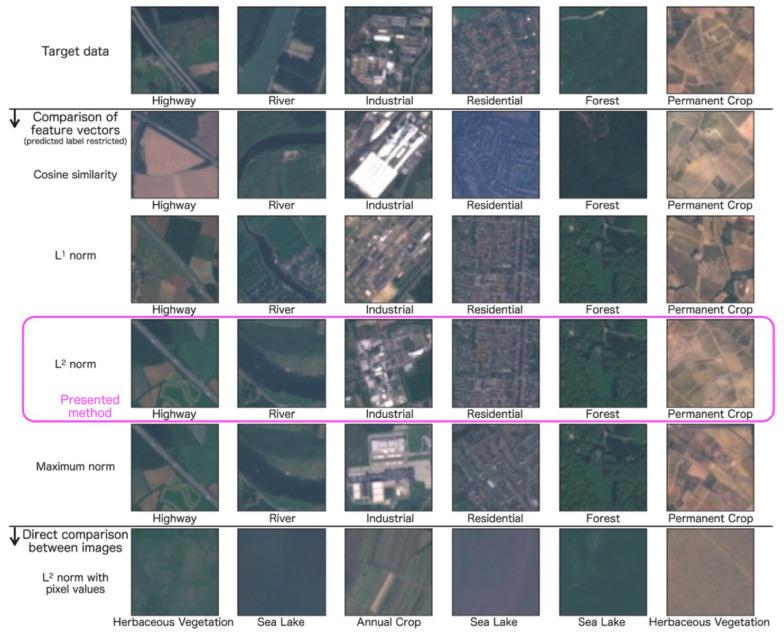
Images selected using the proposed XAI method with different measurement techniques (second to fifth rows) for the input images (first row) and images exhibiting the smallest L2-norm of the pixel values (sixth row). Each image is appropriately labeled. All predicted labels in the displayed images are correct. In the proposed method, the label of the selected image consistently matches that of the original image [176].

**Table 1 sensors-25-05965-t001:** Characteristics, advantages, and disadvantages of SVM, RF, KNN, clustering algorithms, and GBM algorithms.

Algorithm	Characteristics	Advantages	Disadvantages
SVM	Supervised learning algorithm; effective for small datasets; uses a hyperplane to classify data; kernel functions for non-linear problems [99]	High accuracy for small/medium datasets; effective for high-dimensional data; suitable for hyperspectral image classification in RS [100]	Computationally expensive for large datasets; difficult to tune (e.g., choosing kernel and parameters); sensitive to noise and overlapping classes [101]
RF	Ensemble of decision trees; Supervised learning; Uses bagging and feature randomization; Handles categorical and continuous variables [102]	Robust to overfitting; Works well for multispectral/hyperspectral data; Handles missing data well; Easy to interpret variable importance [103]	May struggle with high-dimensional data (e.g., hyperspectral images) without proper tuning; Computationally expensive for large datasets [104]
KNN	Instance-based supervised learning; Non-parametric; Classification based on majority vote of neighbors [105]	Simple and intuitive; Effective for small-scale, low-dimensional RS data; No training phase, fast implementation [106]	Computationally expensive at prediction time; Requires careful selection of k and distance metric; Sensitive to noisy or imbalanced data [107]
Clustering Algorithms (e.g., K-Means, DBSCAN)	Unsupervised learning; Groups data points based on similarity; K-Means: Assumes spherical clusters; DBSCAN: Handles arbitrary shapes [108]	Useful for land-cover mapping and unsupervised classification; K-Means: Fast and easy to implement; DBSCAN: Effective for noisy and non-linear cluster shapes [88]	K-Means: Sensitive to initialization and outliers; DBSCAN: Parameter sensitivity (e.g., epsilon and min points); struggles with high-dimensional data [109]
GBMs and XGBoost	Ensemble of weak learners (decision trees); sequential boosting to reduce error; XGBoost: Optimized, faster implementation of GBM [110]	High predictive accuracy for land-cover classification and change detection; handles missing data well; XGBoost: fast and scalable for large RS datasets [111]	Computationally expensive for large datasets; requires extensive parameter tuning; may overfit if not regularized properly [112]

**Table 3 sensors-25-05965-t003:** Comparative performance of representative AI and ML approaches in RS applications. Reported metrics (e.g., accuracy, RMSE, PSNR) are drawn from the cited studies to illustrate the relative effectiveness of different methods across key tasks. While classical ML methods (e.g., SVM, RF, KNN) remain competitive in certain domains, DL architectures (e.g., CNNs, RNNs, GANs) generally achieve higher accuracy but require larger datasets and greater computational resources.

Study (as Cited)	RS Task	Method(s)	Dataset/Imagery	Reported Performance	Notes/Limitations
Salleh et al. [61]	Urban surface mapping	SVM vs. Rule-based	WorldView-2	SVM: 75.1%; Rule-based: 88.6–92.2% (overall accuracy)	SVM struggled with impervious surfaces and mixed urban objects
Pacheco et al. [78]	Fire-affected area classification	KNN vs. RF	Landsat-8, Sentinel-2, Terra	Accuracy: 89–93%; AUC > 0.88	Spectral mixing and sensor timing caused commission (1–15.7%) and omission (8.8–20%) errors
Han et al. [71,72]	Soil moisture estimation	Physics-informed ML (RF-based)	Multi-source RS + ISMN	RMSE: 0.05 cm^3^/cm^3^; R = 0.9	Robust dataset (1 km resolution), but computationally intensive
Kattenborn et al. [114]	Vegetation species mapping	CNN (U-Net)	UAV RGB imagery	Accuracy ≥ 84%	Relied on high-quality annotated training data
Tang et al. [121]	Land cover classification	DRRNN	5 RS datasets	State-of-the-art (highest accuracy among compared DL methods)	Improved misclassification correction via spatial autocorrelation
Jin et al. [126]	Cloud removal in RS images	HyA-GAN	RICE dataset	Outperformed existing models on PSNR	GAN training unstable, requires careful tuning

**Table 4 sensors-25-05965-t004:** Summary of future prospects for AI in RS, including methodological advances (e.g., multimodal learning, model efficiency, explainability, collaborative frameworks) and application-driven opportunities (e.g., climate and environmental monitoring).

Climate Challenges	AI Solutions
Strengthening Infrastructure	AI enhances the predictive maintenance of infrastructure, including roadways, power systems, and water facilities, while also evaluating new project designs for their vulnerability. Additionally, it supports environmental monitoring and pollution control efforts [188].
Reducing Carbon Emissions	AI-based “energy and carbon footprint modeling” helps track future carbon emissions by analyzing current data, allowing for targeted optimization strategies in industries and transportation, ultimately reducing overall emissions [189].
Managing Vulnerability and Risk	AI utilizes 3D image recognition in UAV-based digital models to assess the seismic vulnerability of buildings in earthquake-prone areas. It also uses artificial neural networks for flood risk assessments, predicting areas most at risk and helping guide preparedness strategies [190].
Preserving Biodiversity	AI-driven technologies, such as image recognition in agricultural vehicles and field robots, assist in identifying and eliminating harmful pests and weeds, administering pesticides in controlled doses to protect ecosystems [191].
Education and Changing Consumer Behavior	AI and ML algorithms leverage consumer data to recommend eco-friendly products and green courses, influencing long-term shifts toward sustainable consumption. For instance, Klimakarl, an AI chatbot, encourages “green behavior” in office settings, promoting climate-conscious decisions [192].
Climate Finance	AI and ML applications in climate finance improve carbon price forecasting, energy cost models, and provide environmental and financial insights that guide both mitigation and adaptation investment decisions [193].
Establishing Early Warning Systems	By integrating AI with wireless sensor networks and IoT, real-time monitoring of factors that trigger landslides or industrial hazards, such as liquid metal leaks, can be achieved, enabling timely warnings and interventions [194].
Anticipating Long-term Climate Change	AI can bolster long-term climate projections by analyzing historical data and creating both regional and global climate models. These models help local authorities assess potential risks from disasters like floods or wildfires and make informed decisions regarding economic and infrastructural planning [195].
Managing Crises	AI-powered hybrid models, integrating multi-criteria decision-making, are useful for flood risk assessment. These models can pinpoint vulnerable areas and preemptively suggest strategies to mitigate damage during crises [196]
Green Economic Recovery	AI accelerates green economic recovery by improving productivity, fostering innovation, and optimizing resource use. Through smart robots and machine vision, AI also helps reduce waste and pollution, making industries more resource-efficient [197]
Climate Research and Modeling	AI’s capabilities in pattern recognition, prediction, and NLP advance climate research by addressing key challenges in reducing climate impacts and enhancing resilience strategies [198].
Collaboration and Partnership	“AI for the Planet” is a collaborative initiative that uses AI to tackle the climate crisis. It encourages global partnerships, advanced analytics, and innovation in AI to develop solutions for sustainable climate actions, while fostering collaboration among stakeholders [199].
Accessibility to Climate Data	The “AI Climate Impact Visualizer” is an interactive tool that allows users to explore AI-generated visualizations of future climate impacts, such as floods or hurricanes, in their areas. It also provides scientific explanations for these predictions, making climate data more accessible to the public [200]

## Data Availability

The data are contained within the article.

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
