# Peer review of "A Comprehensive Review of Remote Sensing and Artificial Intelligence Integration: Advances, Applications, and Challenges"

_sensors, 2025, doi:10.3390/s25195965_

Round 1

Reviewer 1 Report

Comments and Suggestions for Authors

This paper reviewed the developments at the intersection of RS and AI, which is valuable for enhancing the interpretation and processing of RS data. However, the following drawbacks should be taken into consideration.

  1. In Section 3, AI in RS with case studies is presented as a separate review area, while ML in RS and DL in RS are treated as distinct branches. This classification seems inconsistent and conceptually inaccurate. DL is a subfield of AI, and ML is also a research field. In contrast, case studies are tools or examples that span multiple research areas, rather than constituting a separate research branch.
  2. Several figures are reproduced directly from the cited literature, illustrating specific approaches. The reason for citing and reproducing these figures is unclear and needs clarification.
  3. The classification of CNNs in RS under ML in RS is debatable, as CNNs are widely accepted as prominent DL models. The authors should reevaluate or justify this categorization.
  4. Define abbreviations the first time they appear to improve clarity for readers unfamiliar with the terminology.
  5. In Section 4, the authors mention various challenges from the literature and show related results from other studies. It is unclear why these materials were excluded from Section 3, especially since the review methods in both sections appear to be the same (i.e., citing existing work without further synthesis).
  6. In Section 4.2, including climate-related challenges alongside challenges like collaborative frameworks appears conceptually inconsistent and somewhat confusing. The authors might want to reconsider the parallel structure in this subsection.

Reviewer 2 Report

Comments and Suggestions for Authors

dear Authors,

your manuscript titled “Integrating Remote Sensing and Artificial Intelligence: A Comprehensive Survey of Advances and Applications” presents a broad and well-organized review of the integration between remote sensing (RS) and artificial intelligence (AI). You successfully cover a wide range of AI techniques—including machine learning, deep learning, explainable AI, and multimodal fusion—alongside relevant applications in environmental monitoring, agriculture, urban planning, and disaster response. The paper is timely and informative, and it has the potential to serve as a valuable reference for researchers and practitioners. However, several aspects require substantial revision to improve the clarity, scientific rigor, and usefulness of the manuscript.

1)Title Clarification: The current title may suggest a novel methodological contribution. Since the manuscript is a review, I recommend revising the title to explicitly reflect this, e.g.: A Comprehensive Review of Remote Sensing and Artificial Intelligence Integration: Advances, Applications, and Challenges

2) Strengthen Critical Analysis: The manuscript is largely descriptive. A more critical evaluation of the reviewed methods—including their limitations, comparative performance, and suitability for different RS tasks—would enhance its scientific value.

3) Methodological Transparency: Please clarify the criteria used to select the studies included in the review (e.g., databases searched, keywords, inclusion/exclusion criteria). This would improve reproducibility and transparency.

4)Quantitative Synthesis: If feasible, include comparative tables or meta-analyses summarizing performance metrics (e.g., accuracy, precision, recall) across key studies. This would help readers assess the relative effectiveness of different approaches.

5)Ethical and Societal Considerations: The manuscript briefly mentions ethical concerns. A more detailed discussion on issues such as bias, privacy, and the societal impact of AI-driven RS systems would be beneficial.

6)Minor suggestions: 

  • Consider adding visual summaries (e.g., conceptual diagrams, flowcharts) to synthesize key concepts and workflows.
  • Ensure consistent terminology and abbreviation usage throughout the manuscript.
  • A glossary of acronyms could improve readability for interdisciplinary audiences.

Summarizing your manuscript is a strong candidate for publication following major revision. The suggested improvements will help position the paper more clearly as a review, enhance its scientific depth, and increase its utility for the research community.

Reviewer 3 Report

Comments and Suggestions for Authors

The reviewed manuscript (sensors-3768916) presented a hot topic (integration of Remote Sensing and Artificial Intelligence). This review explores how the integration of remote sensing and artificial intelligence can transform Earth observation to enable automated, efficient, and accurate analysis of large and complex datasets? The authors reviewed a lot of relevant literature and presented their review in logical structure, including fundamentals, algorithms (machine learning and deep learning), applications, challenges, and prospects. The review enriched with case studies, which add value. The topic of this manuscript is relevant to the journal's audience and is of considerable value to them. Please consider the following comments:

  1. Keywords; Please reconsider the used keywords, you should add more specific words instead of general words (for example, hyperspectral imaging).
  2. Introduction; Please expand the research gap (why is a new survey necessary?).
  3. Line 49; obtaining → extracting.
  4. Lines 128:145; There is no need to all the details related to the study area in the presented case study (Stockholm), only focus on the methods and the main finding that support your section (Fundamentals of RS). In addition, I think the presented figure (Figure 1) is not necessary.
  5. Figure 1; Labels in this figure are not clear.
  6. Sections 3.2. and 3.3.; Please reduce algorithm explanations, as much as you can, and instead focus on applications and performance comparisons in RS (No need for basic information, this is not a textbook chapter).
  7. Line 245; convincing → powerful or effective.
  8. Line 273; “Methodological framework [61]” for what? Please modify this figure caption with more details.
  9. Line 355; Inspired by → Adapted from or Modified after, …(Same for Figure 11).
  10. Line 399; Please modify this figure caption with more details.
  11. Line 471; managing → handling or processing.
  12. Line 662; occurred → occurred.
  13. Please add a subsection discussing “ethical considerations” as mentioned in the abstract.

Comments on the Quality of English Language

The manuscript contains a lot of very long sentences, and in some cases, sentences are cut off.

Round 2

Reviewer 1 Report

Comments and Suggestions for Authors
  1. Although the authors explained why they included AI in RS in Section 3, I don't think the reason aligns with the structure of this section. Changing the subsection's title cannot resolve the logical issue caused by this subsection.
  2. Except for comment 4, the authors did not provide meaningful revisions based on the other comments. They added some explanations of their reasoning, but I cannot agree with them.

Reviewer 2 Report

Comments and Suggestions for Authors

Dear authors, you incorporated all the main observations in a timely and convincing manner.
In particular:

Title: has been changed to clarify that this is a review, as requested.
Critical analysis: analytical steps have been strengthened, with obvious improvements in sections 3 and 4. Tables 2 and 3 now include quantitative comparisons between methods (accuracy, RMSE, PSNR, etc.).

Methodological transparency: a detailed description of study selection criteria (database, keywords, inclusion/exclusion criteria) has been added.

Quantitative summary: the new Table 3 is well done and useful for comparisons between approaches.

Ethical and social aspects: a dedicated subsection has been added, with references to bias, privacy and social impacts.

Minor improvements: diagrams have been added (e.g., Fig. 11), terminology standardized, and a glossary inserted.

Your manuscript is now acceptable for publication. The changes made are substantial, well justified, and well documented in the response letter. The scientific quality and clarity of the contribution have improved significantly.

Reviewer 3 Report

Comments and Suggestions for Authors

The authors' improvements are satisfactory. The reviewer expresses gratitude to the authors for replying to all comments. One point is not addressed:

Line 364; Inspired by → Adapted from or Modified after, …(Same for Figure 11). Please remember to change it during proofreading.
